

# Global flux-based ozone risk assessment for wheat up to 2100 under different climate scenarios

Pierluigi R. Guaita[1,8], Riccardo Marzuoli[1], Leiming Zhang[2], Steven Turnock[3,4], Gerbrand Koren[5], Oliver Wild[6], Paola Crippa[7], Giacomo Gerosa[1]

[1]Dep. Mathematics and Physics, Catholic University of the Sacred Heart, Brescia, Italy
[2]Air Quality Research Division, Science and Technology Branch, Environment and Climate Change Canada, Toronto, Canada
[3]Met Office Hadley Center, Exeter, UK
[4]University of Leeds Met Office Strategic (LUMOS) Research Group, University of Leeds, UK
[5]Copernicus Institute of Sustainable Development, Utrecht University, Utrecht, The Netherlands
[6]Lancaster Environment Centre, Lancaster University, Lancaster, UK
[7]Department of Civil and Environmental Engineering and Earth Sciences, University of Notre Dame, Notre Dame, IN, USA
[8]Department of Applied Computational Mathematics and Statistics, University of Notre Dame, Notre Dame, IN, USA

*Correspondence to*: Pierluigi R. Guaita (pierluigirenan.guaita@unicatt.it), Giacomo A. Gerosa (giacomo.gerosa@unicatt.it)

**Abstract.** The negative effects of tropospheric ozone ($O_3$) on vegetation can lead to reduced photosynthesis, accelerated leaf senescence, and other negative outcomes which affect crop yields and biodiversity. This study presents a flux-based assessment of the global impact of $O_3$ on bread wheat (*Triticum aestivum*) for the 21$^{st}$ century, under various climate scenarios (Shared Socioeconomic Pathways, SSPs). A dual-sink big-leaf dry deposition model is employed to estimate the phytotoxic ozone dose (POD) absorbed by wheat through stomata, integrating data from two Earth System Models (ESMs)
from the Coupled Model Intercomparison Project 6 (CMIP6). The study explores spatial and temporal variations in $O_3$ concentrations and the effects of climate variables on stomatal conductance, explaining changes in POD from the present time to the century's end. The results indicate significant regional disparities in $O_3$ dose for wheat, particularly under weak $O_3$ precursor emissions control scenarios. The most vulnerable regions include Northern Europe, East China, and the Southern and Eastern edges of the Tibetan Plateau, where the POD increase by the end of the century is expected to be most
pronounced. Conversely, POD decreases worldwide under stringent pollution emission control scenarios. However, in some regions, changes in POD may be driven more by climate variables and their interaction with $O_3$, rather than by $O_3$ concentrations alone. Therefore, this study emphasizes the need for effective emission mitigation policies of both $O_3$ precursors and greenhouse gases to preserve global food security from $O_3$ damages.

## 1. Introduction

Tropospheric ozone ($O_3$) is a highly reactive gas that can harm vegetation and ecosystems. $O_3$ damages vegetation by altering plant biochemistry and physiology after entering leaf stomata, resulting in reduced photosynthesis, accelerated leaf senescence, and causing the expression of detoxification systems. This in turn reduces canopy carbon gain and export, yield,





and biodiversity, among other negative effects (Fuhrer et al., 2016; Grulke and Heath, 2020; Ramya et al., 2023; Wright et al., 2018). Vegetation impacts have consequences for food security, as highlighted by (Emberson, 2020).

$O_3$ impacts on vegetation have been quantified primarily via two methods: the exposure (concentration)-based or the dose (flux)-based approach. Comparisons of the two approaches have been performed over different vegetation types (Anav et al., 2016; Hoshika et al., 2020; Karlsson et al., 2007; Mao et al., 2024; Mills et al., 2018; Paoletti et al., 2019; Pleijel et al., 2022; Simpson et al., 2007; Tai et al., 2021; Tang et al., 2014). The flux-based approach, which estimates the phytotoxic ozone dose (POD), is more biologically meaningful to estimate $O_3$ damage to vegetation, as it accounts for the actual amount of $O_3$
entering the plants through the stomata, i.e. the stomatal flux, rather than on the simple $O_3$ concentration surrounding the plants (Paoletti and Manning, 2007). Existing schemes for estimating $O_3$ stomatal flux to vegetation vary in complexity, e.g., as a simple function of temperature and solar radiation (Wesely, 1989), a single leaf Jarvis model (Baldocchi et al., 1987; Jarvis, 1976), a sunlit/shade (two-big-leaf) scheme (Emberson et al., 2000; Zhang et al., 2003), a photosynthesis approach (Ball et al., 1987; Charusombat et al., 2010), and various combinations of the above (Clifton et al., 2023; and references
therein).

Regional- to global-scale $O_3$ risk assessments for various vegetation types have been extensively conducted over the past two decades, using these schemes or exposure-based approaches, and estimating past and present $O_3$ damage over different canopies (Anav et al., 2011; Cheesman et al., 2023; Guaita et al., 2023; Mills et al., 2011; Savi et al., 2020; Sharps et al., 2021a). Many studies concluded that high $O_3$ concentrations can cause significant crop losses and economic damages
worldwide. For example, for present-day scenarios, Pleijel et al. (2018) indicated ~8% reduction in wheat production in Europe, North America and Asia, Tai et al. (2021) estimated globally aggregated yield losses of up to 7% for four staple crops, and even higher percentage losses have been reported by other studies as reviewed by Emberson (2020).

Estimation of future $O_3$ impacts on vegetation have been mostly based on the exposure-based approach (e.g., Chuwah et al., 2015; Sicard et al., 2017). The few studies that have explicitly made regional flux-based estimates of future $O_3$ impacts, have
assumed either present-day meteorology with projected emissions, or present-day emissions with future climate (Klingberg et al., 2014; Simpson et al., 2007; Tang et al., 2014). To our knowledge, a comprehensive, global flux-based analysis of future $O_3$ impacts across the 21$^{st}$ century, considering both future climate and emission scenarios, is still lacking.

The present study was designed within the framework of the Ozone Deposition Focus working group of the Tropospheric Ozone Assessment Report initiative, Phase II (TOAR-II), to make a global-scale flux-based assessment of future $O_3$ risks for
bread wheat (*Triticum aestivum*), which is one of the world's most important staple food crops and often regarded as a reference for species with high $O_3$-sensitivity (Mills et al., 2011; Sitch et al., 2007). More specifically, the objectives of this research are:

1. To estimate future trends of POD for bread wheat to the end of the 21$^{st}$ century under different climate change scenarios.
2. To identify vulnerable regions to future food security threats due to the negative $O_3$ effects on wheat.
3. To inform adaptation and mitigation policies to minimize future $O_3$ risk.





To address these objectives, we employ a dual-sink big-leaf dry deposition model (Guaita et al., 2023), using meteorology and $O_3$ concentrations inputs from Earth System Models (ESMs) participating in the Coupled Model Intercomparison Project 6 (CMIP6; Eyring et al., 2016). To generate a set of future climate simulations, these ESMs use the forcing datasets associated with the so-called "shared socioeconomic pathways" (SSP; Riahi et al., 2017), which are future pathways combining different trends in social, economic and environmental developments with different assumptions about anthropogenic emission mitigation applied on top of these to meet pre-defined climate targets (radiative forcing).

In the following sections, we describe the adopted methodology, including the analyzed CMIP6 models and the dry deposition model used to calculate the POD. We present the main results – i.e. POD trends and maps – and we quantify the interaction of $O_3$ risk with climate and $O_3$ concentrations. Finally, we discuss the confidence in our results, contextualizing them in the broader SSP framework.

## 2. Methodology

This section describes the CMIP6 models and experiments involved in this study, along with associated information on sowing dates, plant available water and global region definitions used in this study. Then, we summarize the main features of the $O_3$ dry deposition model for wheat (Guaita et al., 2023) that is employed to calculate the POD metric, as well as the parameterization of the stomatal conductance module. Finally, we outline the methods adopted to present the model outputs and describe the statistical tools used for the interpretation of the results.

### 2.1 Selection of CMIP6 models and SSPs

A subset of models from the CMIP6 experiment is selected as input to this work. Specifically, we apply the following criteria to identify the model runs suitable for our study: (i) an online/coupled-chemistry framework (AerChemMIP; Collins et al., 2017) to include the feedback of $O_3$ on climate and (ii) sub-daily temporal resolution of meteorological variables to enable $O_3$ flux calculations. According to these criteria, we identify GFDL-ESM4 (Dunne et al., 2020) and UKESM1-0-LL (Sellar et al., 2019) for this study. UKESM1-0-LL and GFDL-ESM4 are both fully-coupled global ESMs, which include a physical atmosphere-ocean model coupled with additional interactive earth system components including; ocean biogeochemistry, stratosphere-troposphere chemistry and aerosol scheme and terrestrial carbon cycles coupled to interactive vegetation (Table 1). They both have a horizontal grid resolution of between 100 to 140 km in the mid-latitudes, with vertical levels extending to the upper stratosphere. Comprehensive chemistry schemes are included within both ESMs simulating the reactions and transport of the major chemical species (odd-oxygen $O_x$, nitrogen $NO_y$, hydrogen $HO_x = OH + HO_2$, carbon monoxide CO, methane and short-chain non-methane volatile organic compounds) involved in ozone formation. In this way the ESMs are able to simulate the interaction between changes in climate and chemistry in both the historical period and future scenarios.



Output from GFDL-ESM4 and UKESM1-0-LL simulations is publicly available from the ESGF metagrid site (Cinquini et al., 2014). The model variables required for this study are as follows: surface upward sensible heat flux (*hfss*, in ESGF Metagrid notation), near-surface specific humidity (*huss*), surface downwelling/upwelling longwave/shortwave radiation

(*rlds*, *rlus*, *rsds*, *rsus*, respectively), precipitation (*pr*), surface air pressure (*ps*), near-surface wind speed (*sfcWind*), and near-surface air temperature (*tas*). O$_3$ concentrations at the lowest model level (sfo3) are requested to be provided at an hourly resolution by models participating in AerChemMIP. Additionally, 3D air temperature and specific humidity (*ta* and *hus*, respectively) are used to convert model level height (Pa) of GFDL-ESM4 to geometric height (m above ground), which is a requirement of the dry deposition model used in this study (Guaita et al., 2023). On the other hand, UKESM1-0-LL uses a

hybrid-height coordinate system, and therefore such conversion is not required for this model. See Appendix for details.

**Table 1: Key features of the GFDL-ESM4 and UKESM1-0-LL runs used in this study: horizontal nominal resolution, vertical atmosphere resolution, atmosphere and chemistry modules wth the corresponding references, variant indicating configurations (r=realization of the ensemble, i=initialization method, p=physics version, f=forcing), output frequency (hfss=surface upward**
**sensible heat flux, hus= specific humidity, huss=near-surface specific humidity, rlds=surface downwelling longwave radiation, rlus =surface upwelling longwave radiation, rsds=surface downwelling shortwave radiation , rsus=surface upwelling shortwave radiation, pr=precipitation, ps=surface air pressure, sfcWind=near-surface wind speed, sfo3=O$_3$ concentrations at the lowest model level, ta=air temperature and tas=near-surface air temperature), and citations for documentation and datasets.**

|  |  | GFDL-ESM4 | UKESM1-0-LL |
|---|---|---|---|
| Horizontal resolution (lon × lat) |  |  |  |
|  | Land | 1°×1.250° | 1.250° × 1.875° |
|  | Ocean | 0.5° tripolar | 1° tripolar |
| Vertical atmosphere resolution |  | 49 levels to 1Pa (~80km) | 85 levels to ~85km |
| Atmosphere module |  | GFDL-AM4.1 (Horowitz et al., 2020) | HadGEM3-GC3.1 (Kuhlbrodt et al., 2018; Williams et al., 2018) |
| Chemistry module |  | GFDL-ATMCHEM4.1 (Horowitz et al., 2020) | UKCA-StratTrop (Archibald et al., 2020; Mulcahy et al., 2018) |
| Variant |  | r1i1p1f1 | r1i1p1f2 |
| Output frequency | 1 hour | ps[a], tas[a], sfo3 | ps, sfo3 |
|  | 3 hours | ps[a], tas[a], huss, rlds, rlus, rsds, rsus, pr | rlds, rsds, rsus, pr, tas, hfss |
|  | 1 day | sfcWind | huss, rlus, sfcWind |



| | 1 month | hfss, hus, ta | |
|---|---|---|---|
| Documentation | | Dunne et al. (2020) | Sellar et al. (2019) |
| Dataset citation | | Horowitz et al. (2018); Krasting et al. (2018) | O'Connor (2020); Tang et al. (2019) |

**[a]ps and tas output frequencies are 1 hour for the historical experiment and SSP3-7.0, 3 hours for SSP1-2.6 and SSP5-8.5.**

$O_3$ risk for wheat cultivation up to 2100 is quantified with respect to a POD baseline value, which is calculated as the average for the period 2000-2014. The POD over the baseline years is derived from the 'historical' experiment of CMIP6 (Eyring et al., 2016, p.201), which is an experiment performed by every CMIP6 model using standardized input data. Future $O_3$ risk is estimated based on climate and emission scenarios described by the SSPs (Riahi et al., 2017), which were developed for the CMIP6, and results were used in the IPCC Sixth Assessment Report (Intergovernmental Panel On Climate

Change (IPCC), 2023). Specifically, we focus on the scenarios SSP1-2.6 (Van Vuuren et al., 2017), SSP3-7.0 (Fujimori et al., 2017), and SSP5-8.5 (Kriegler et al., 2017), as they represent contrasting characteristics in terms of future climate change and emission policies for air quality that impact $O_3$ concentrations from 2015 to 2100. Herein we classify SSP1-2.6 as a low-emissions and low radiative forcing scenario, SSP3-7.0 as a high-emissions and high radiative forcing scenario, while SSP5-8.5 as a high radiative forcing and partial emission control scenario, with controls beginning in the second half of the 21st

Century (see Table 2). Thus, from an $O_3$ concentration perspective, SSP5-8.5 can be broadly considered as an intermediate scenario, with end-of-century pronounced climate change and $O_3$ concentrations akin to those observed in the historical baseline (Turnock et al., 2020). Simulations for these SSPs are available from both GFDL-ESM4 and UKESM1-0-LL include the aforementioned SSPs. Further, the SSP3-7.0pdSST experiment (present-day Sea Surface Temperature; Zanis et al., 2022) is available from UKESM1-0-LL and is included in this study to assess the effect of changing $O_3$ concentrations

only in a high-emission/present-day climate scenario.

**2.2 Sowing dates and soil hydraulic properties**

The $O_3$ stomatal flux model requires wheat coverage and annual sowing date maps to simulate plant growth. Qiao et al. (2023) produced a global dataset of sowing dates and wheat coverage for *T. aestivum* averaged over specific multi-year periods (1990-2000, 2020-2029, 2040-2049, 2090-2099) with a resolution of $0.5° \times 0.5°$ for SSP1-2.6 and SSP3-7.0. In this

work, these dates are linearly interpolated to obtain yearly maps of sowing dates for the whole globe. The results obtained for SSP3-7.0 are also used for SSP5-8.5, as the two scenarios have comparable radiative forcing. This dataset is upscaled to the resolution of the CMIP6 models.

Contextually, $O_3$ stomatal flux model requires also wilting point and field capacity soil maps, to simulate plant available water during the growing season. Wilting point is defined as the minimum amount of water in the soil that plants require not

to wilt. On the contrary, field capacity is the maximum amount of water contained in the soil after it has been thoroughly saturated and allowed to drain. Both of them are expressed as $m^3$ of water to $m^3$ of soil and they vary according to soil



texture. Zhang et al. (2018) developed maps of wilting point and field capacity with a resolution of $1 \times 1$ km². The value of field capacity at each CMIP grid node is obtained by averaging all the $1 \times 1$ km² nodes provided by Zhang et al. (2018) that belong to the nodes that Qiao et al. (2023) indicated as wheat area. The same is done for the wilting point maps.

## 2.3 Dry deposition model for wheat

In this work we apply the $O_3$ dry deposition model developed by Guaita et al. (2023) to compute $POD_6$ values over the entire globe. This model is a dual-sink big-leaf model that simulates crop geometry, phenology, plant available water in the soil, light penetration within the canopy, and calculates stomatal conductance ($g_s$) and $O_3$ uptake by plants, following the DO3SE paradigm originally developed by Emberson et al. (2000). The $O_3$ stomatal flux model receives as input meteorological and chemical data from the CMIP6 models under the different SSPs. The model code is developed in MATLAB (version R2023a).

$O_3$ deposition from the lowest model level to the vegetated surface is calculated with a resistive network of three resistances in series: an atmospheric resistance $R_a$ representing turbulent mixing from the lowest model level to near-surface height, a quasi-laminar sublayer resistance $R_b$ representing diffusive transport to the surface, and a surface resistance $R_c$. The latter consists of three resistances in parallel: a cuticular resistance $R_{cut}$, a stomatal resistance $R_{stom}$ and a ground resistance composed of an intra-canopy resistance $R_{inc}$ and a soil resistance $R_{soil}$, in series. $R_a$ is calculated following the assumptions of the Monin-Obukhov similarity theory, $R_b$ is calculated with the formulation of Hicks et al. (1989), $R_{cut}$ and $R_{soil}$ are assumed as constant values of 1500 and 200 s m⁻¹ respectively, and $R_{inc}$ was calculated following the formulation of Erisman et al. (1994).

The $R_{stom}$ was calculated as the inverse of $g_s$ by applying the empirical Jarvis-Stewart approach (Jarvis, 1976; Stewart, 1988), which is based on the limitation effect of the main environmental variables on the maximum stomatal conductance to water ($g_{max}$) of wheat:

$$g_s = g_{max} \cdot min\{f_{phen}, f_{O3}\} \cdot f_{light} \cdot max\{f_{min}, f_{temp} \cdot f_{VPD} \cdot f_{soil}\} \tag{1}$$

where the $f$ functions (all ranging between 0 and 1) describe the limiting effect on $g_{max}$ due to environmental variables such as light ($f_{light}$), temperature ($f_{temp}$), air water Vapour Pressure Deficit ($f_{VPD}$) and plant available water in the soil ($f_{soil}$), and to phenological growth ($f_{phen}$) and $O_3$ dose received by the plants ($f_{O3}$). The $f_{min}$ term represents a constant value of 0.01 indicating the minimum $g_s$ expressed relative to $g_{max}$ during daylight hours. For this study an $f_{clim}$ term is defined, which represents the product $(f_{temp} \cdot f_{VPD} \cdot f_{soil})$ and combines the effect of the main climatic factors on $g_s$.

The $O_3$ dry deposition model considers both sunlit and shaded leaves and varies the crop geometry according to the growth and phenology of wheat plants. Further details on the model can be found in Guaita et al. (2023).

For the simulations of this work, different parameterizations of $g_s$ for *T. aestivum* are adopted depending on the different biogeographical regions. Contextually to sowing dates, Qiao et al., (2023) categorized wheat-growing areas into four climatic zones (temperate, cold, warm, and monsoon) based on the intensity and seasonality of temperature and precipitation.



According to this classification, we use the following wheat parameterizations of the $g_s$ model, depending on the climatic zone: spring wheat (Grünhage et al., 2012) in the cold region, winter wheat (Grünhage et al., 2012) in the temperate region, and mediterranean wheat (González-Fernández et al., 2013) in the warm and monsoon regions. The only difference between the spring wheat and winter wheat parameterization is in the $f_{phen}$ limiting function, which considers the different temperature sums used to calculate the mid-anthesis date (the middle of the flowering period of a plant) and the corresponding flux accumulation period.

The $g_s$ model for the Mediterranean region was originally parameterized based on field $g_s$ measurements collected in Spain (González-Fernández et al., 2013, currently adopted in the Mapping Manual of the Convention of Long Range Transport of Air Pollution of the UN Economic Convention for Europe, LRTAP Convention, 2017). In this study, however, it is also used for warm and monsoon climatic zones. Although parameterizations exist for specific regions within these zones, these were found to be incomplete for the application in our study (Feng et al., 2012) or very similar to the one by González-Fernández et al. (2013) (Yadav et al., 2021).

Once $g_s$ is obtained, the seasonal $O_3$ stomatal dose received by wheat plants can be calculated as the phytotoxic $O_3$ dose above the $Y$ threshold accounting for plants detoxification capacity ($POD_Y$; mmol $O_3$ m$^{-2}$ of projected leaf area) by integrating the hourly $O_3$ stomatal flux ($F_{sO3,i}$; nmol $O_3$ m$^{-2}$ s$^{-1}$) over the flux accumulation period ($i = 0, \dots t$, i-th hour of the accumulation period):

$$POD_Y = \sum_{i=0}^{t} \max\{F_{sO3,i} - Y, 0\} \cdot 3600 \cdot 10^{-6} \qquad (2)$$

where the $F_{sO3,i}$ is obtained from the $O_3$ concentration at top canopy height (see Eq. 51, 52 in the Appendix of Guaita et al., 2023b). For *T. aestivum* a detoxifying threshold of 6 nmol $O_3$ m$^{-2}$ s$^{-1}$ ($POD_6$) is recommended by the (LRTAP Convention, 2017), while the accumulation period for $POD_6$ calculation runs from the beginning of the anthesis to the plant's maturity, which depends on the phenology limiting function ($f_{phen}$). In our case, plant's maturity is set at 1775, 2325 and 2400 °C day for spring wheat, winter wheat and Mediterranean wheat, respectively (González-Fernández et al., 2013; Grünhage et al., 2012).

A well-established dose-response relationship for *T. aestivum* based on $POD_6$ and relative grain yield (Grünhage et al., 2012; LRTAP Convention, 2017; Pleijel et al., 2007) can be used to calculate the relative yield loss (RYL; %):

$$RYL = 3.85 \cdot POD_6 \qquad (3)$$

This relationship predicts a 5% decrease of grain yield for each increment of $O_3$ dose of 1.3 mmol m$^{-2}$ (PLA, plant leaf area), and this is defined as the 'Critical Level' (CL).

Compared to the original model described in Guaita et al. (2023), in this exercise (i) pressure and sensible heat flux are provided as CMIP6 input, (ii) vapor pressure is calculated from specific humidity by $e$ [kPa] $= p \cdot q/0.622$, with $p$ air pressure [kPa], and $q$ the specific humidity [kg $H_2O$ (kg air)$^{-1}$], and (iii) net radiation is calculated by summing the radiation components provided by the CMIP6 models.





**2.4 Assumptions and description of dry deposition model runs**

The $O_3$ stomatal flux model uses output from the ESMs at their native spatial resolution and runs at an hourly timestep. However, the CMIP6 models a have different output frequency (Table 1) and, to make full use of the the hourly $O_3$ concentrations provided by the CMIP6 models, other variables at coarser resolutions are interpolated in time by nearest

neighbor interpolation, which, compared to linear interpolation, avoids excessive underestimation of the diurnal peaks. Any output produced in this study has the same spatial resolution as the input data from the CMIP6 model used for the simulations.

Global annual $POD_6$ maps from 2000 to 2099 are produced for each SSP considered. The maps produced are masked with the land-use maps of Qiao et al. (2023). Every node where wheat does not reach maturity before the next prescribed sowing

date (according to González-Fernández et al., 2013 and Grünhage et al., 2012) is excluded from the $POD_6$ map for that year. The model assumes that soil water content is at field capacity at the sowing date every year. Then, to evaluate the effect of soil water on $O_3$ uptake, two contrasting runs are presented: (i) a run that dynamically simulates soil water availability to plants, with precipitation as the only source of water (henceforth rain-fed run), and (ii) a run that removes any limitation due to soil water content, by assuming plant available water at field capacity, i.e. with $f_{soil} = 1$ at every timestep (henceforth FC

run).

**2.5 ANOVA**

A two-way ANalysis Of VAriance (ANOVA; Table 2; von Storch & Zwiers, 1999) is applied to the POD6 values computed at the end of the century in order to assess the areas of the globe where the changes of $POD_6$ with respect to the baseline period are statistically significant. We consider radiative forcing and control emission policies as the two factors, each one

applied at two levels, i.e. low/high for radiative forcing (RF), and weak/strong for emission policy (EP). These levels are determined by the different SSPs, as described by Table 2.

ANOVA was performed only on the $POD_6$ output produced through the UKESM1-0-LL, since the GFDL-ESM4 is incomplete, i.e. does not include the SSP3-7.0pdSST experiments. ANOVA is performed independently in every grid node of the map with $POD_6 > 0$ and a Bonferroni correction is applied to the p-values to account for multiple tests and for multiple

locations ( #tests × #(nodes with $POD_6 > 0$) = $(8 \cdot 7/2)$ × 2751 = 77028 ). Furthermore, this analysis allows to calculate the fraction of explained total variance by each factor, defined as $R^2$, according to the definition by von Storch and Zwiers, 1999. The ANOVA analysis was performed with MATLAB.

**Table 2: Configuration for ANOVA by SSPs, classified by emission control policies (EP) and radiative forcing (RF).**

| | Emission control policies (EP) | |
|---|---|---|
| | Strong | Weak |





| Radiative forcing (RF) | Low | SSP1-2.6 | SSP3-7.0pdSST |
|---|---|---|---|
| | High | SSP5-8.5 | SSP3-7.0 |

## 3. Results

### 3.1 Spatio-temporal patterns of O₃ and climate variables and their impacts on stomatal conductance

Figure 1 shows the $O_3$ mean concentration during the *baseline* years (2000-2014), and its change ($\Delta O_3$) at 2100. $O_3$ concentrations are expressed at the wheat canopy height (see Eq. 51 in the Appendix of Guaita et al., 2023) and averaged over the accumulation period of the stomatal flux. During the baseline years, both GFDL-ESM4 and UKESM1-0-LL show similar global $O_3$ mean concentration and standard deviation (Table 3). The Northern Hemisphere presents higher $O_3$ concentrations than the Southern Hemisphere, with the Middle East and Asia the regions with the highest $O_3$ around the globe (Figure 1a,b). Nevertheless, UKESM1-0-LL shows stronger latitudinal and elevation gradients compared to GFDL-ESM4, and generally reports higher concentrations than the latter for latitudes <45°, with the only exception of the Eastern Indian peninsula. Turnock et al. (2020) showed that all CMIP6 models tend to overestimate climatological surface ozone concentrations in the 2005 to 2014 when compared to observations. The simulated difference of ozone to observations and also between models could be due to a number of different reasons including uncertainties in the input datasets (emission inventories), output processes (deposition) or vertical mixing (Wild et al., 2020). UKESM1-0-LL simulates the strongest seasonal cycle in Northern Hemisphere surface ozone out of all of the CMIP6 models, potentially due to excessive $NO_x$ titration of ozone in this model (Turnock et al., 2020).

Despite these differences, the two models show similar changes across different SSPs. SSP1-2.6 – which assumes the implementation of strong policies to reduce air pollution – indicates a decrease in $O_3$ concentrations by the end of the century for both models (Figure 1c,d). However, the global mean $O_3$ reduction is 3.1 ppb greater in GFDL-ESM4 than in UKESM1-0-LL. On the contrary, under scenario SSP3-7.0, which is characterized by weaker air pollution control policies, $O_3$ concentrations generally increase by 2100, even though some parts of Europe and eastern U.S. show similar or lower averages than the baseline (Figure 1e,f). SSP5-8.5 shows the same spatial patterns as SSP3-7.0, but with lower $O_3$ values at the end of the century, which is consistent with the implementation of strong air quality control policies from 2050 onward, despite the strong climate change signal (Figure 1g,h).

The limiting function $f_{clim}$ combines the effect of the main climatic parameters (air temperature, VPD and plant available water) on stomatal conductance. $f_{clim} \approx 1$ indicates very little limitation to the $g_s$, while descreases in $f_{clim}$ towards zero denote a progressive stomatal closure associated with a decrease in $O_3$ uptake. During the baseline years, the $f_{clim}$ over the accumulation period is greater than 0.5 over large areas of Eastern North America, Europe, and East Asia, while widely limiting conditions ($f_{clim} < 0.2$) are mostly in arid regions of the globe (Figure 2a,b). Figure 2c-h shows the changes in $f_{clim}$



at 2100 compared to the baseline ($\Delta f_{clim}$). From an $O_3$ risk perspective, negative $\Delta f_{clim}$ values (in blue) correspond to a stronger limitation of $g_s$ and thus, potentially, lower $POD_6$ and $O_3$ damage. On the contrary, positive $\Delta f_{clim}$ values (in red)

correspond to a weaker limitation of $g_s$ and higher $POD_6$ and $O_3$ damage. SSP3-7.0 and SSP5-8.5 indicate overall similar magnitude of $\Delta f_{clim}$ across the globe at the end of the century, while SSP1-2.6 shows weaker changes, as expected from the weaker RF of this scenario. However, some regional differences across models can be identified. For instance, a higher $\Delta f_{clim}$ in GFDL-ESM4 is observed over the U.S. and Europe, and across the whole Asia in UKESM1-0-LL. The only consistent pattern in 2100 changes across both models and all SSPs is the large positive shift at latitudes greater than 45°,

which indicates higher stomatal conductance (and better growing conditions) for wheat at higher latitude. This is due to increased $f_{temp}$ values, i.e. temperatures closer to the optimum for stomatal conductance (Figure 3a-d). The function $f_{VPD}$ (Figure 3e-h) has non-limiting conditions ($f_{VPD} > 0.9$) over most of the globe, with the only exception in desert regions, indicating that $f_{clim}$ is only marginally affected by this component. This function becomes generally more limiting at the end of the century, especially for UKESM1-0-LL, because of a general decrease in relative humidity, which is a feature

frequently observed in future climate simulations (e.g., Fang et al., 2022). The spatial patterns of $f_{clim}$ appear to be largely affected by $f_{soil}$, because where $f_{soil}$ is limiting stomatal conductance, $f_{clim}$ becomes also limiting (Figure 3i-l). $f_{soil}$ grows considerably at the end of the century over arid regions, especially for the SSP3-7.0 and the SSP5-8.5. Maps for SSP1-2.6 and SSP5-8.5 may be found in the appendix (Figure A1-3).

**Table 3: Global means (±SD) for $O_3$ concentrations and Jarvis functions, during the baseline period and at 2100 over wheat-growing regions, for two climate models (GFDL-ESM4 and UKESM1-0-LL) under three different SSP scenarios (SSP1-2.6, SSP3-7.0 and SSP5-8.5).**

| Variable | Model | Baseline[a] | 2100[a] | | |
|---|---|---|---|---|---|
| | | | SSP1-2.6 | SSP3-7.0 | SSP5-8.5 |
| $O_3$ [ppb] | GFDL-ESM4 | 35.0 ± 9.1 | 23.7 ± 7.6 | 38.1 ± 12.0 | 36.2 ± 10.9 |
| | UKESM1-0-LL | 36.3 ± 10.4 | 28.1 ± 8.3 | 39.2 ± 10.7 | 37.3 ± 9.4 |
| $f_{clim}$ [0,1] | GFDL-ESM4 | 0.27 ± 0.24 | 0.30 ± 0.24 | 0.35 ± 0.26 | 0.36 ± 0.26 |
| | UKESM1-0-LL | 0.30 ± 0.23 | 0.32 ± 0.24 | 0.37 ± 0.23 | 0.34 ± 0.24 |
| $f_{temp}$ [0,1] | GFDL-ESM4 | 0.58 ± 0.27 | 0.62 ± 0.25 | 0.71 ± 0.19 | 0.71 ± 0.17 |
| | UKESM1-0-LL | 0.58 ± 0.21 | 0.68 ± 0.18 | 0.68 ± 0.17 | 0.71 ± 0.14 |
| $f_{VPD}$ [0,1] | GFDL-ESM4 | 0.89 ± 0.16 | 0.89 ± 0.16 | 0.86 ± 0.18 | 0.85 ± 0.19 |





|  |  |  |  |  |  |
|---|---|---|---|---|---|
|  | UKESM1-0-LL | $0.89 \pm 0.14$ | $0.84 \pm 0.17$ | $0.81 \pm 0.20$ | $0.76 \pm 0.22$ |
|  | GFDL-ESM4 | $0.60 \pm 0.37$ | $0.60 \pm 0.36$ | $0.59 \pm 0.35$ | $0.60 \pm 0.35$ |
| $f_{soil}$ [0,1] | UKESM1-0-LL | $0.62 \pm 0.36$ | $0.57 \pm 0.36$ | $0.69 \pm 0.34$ | $0.64 \pm 0.36$ |

[a]Baseline indicates the mean between 2000 and 2014, and 2100 indicates the mean between 2090-2099.





**Figure 1: Ozone mean concentrations over the baseline period (a,b), and ΔO₃ at 2100 with respect to the baseline across the different SSPs for GFDL-ESM4 (c,e,g) and UKESM1-0-LL (d,f,h). O₃ values are expressed at canopy height and over the POD₆ accumulation period. Baseline indicates the mean between 2000 and 2014, and 2100 indicates the mean between 2090-2099.**







**Figure 2:** $f_{clim}$ mean values over the baseline period (a,b), and $\Delta f_{clim}$ at 2100 with respect to the baseline across the different SSPs for GFDL-ESM4 (c,e,g) and UKESM1-0-LL (d,f,h). $f_{clim} = f_{temp} \cdot f_{VPD} \cdot f_{soil}$ summarizes the limitations to stomatal conductance due to temperature, VPD and plant available water, with values ranging from 0 to 1 depending on whether they are limiting or not. Positive or negative $\Delta f_{clim}$ correspond to higher and lower $O_3$ risk, respectively (red and blue, in c-h). Baseline indicates the mean between 2000 and 2014, and 2100 indicates the mean between 2090-2099.





**Figure 3: Jarvis limiting functions for $g_s$ over the baseline period and the changes at 2100 with respect to the baseline under SSP3-7.0 for $f_{temp}$ (a-d), $f_{VPD}$ (e-h) and $f_{soil}$ (i-l), for GFDL-ESM4 (a,c,e,g,i,k) and UKESM1-0-LL (b,d,f,h,j,l). Baseline indicates the mean between 2000 and 2014, and 2100 indicates the mean between 2090-2099.**



## 3.2 POD$_6$ trends by region

Figure 4 displays yearly POD$_6$ values averaged over different geographical regions under different SSPs. These values are
calculated individually for each model (thin lines), and for each region (different panels). In this analysis, wheat areas with
potential O$_3$ risk are identified considering only the nodes where POD$_6$ was greater than 0.65 mmol m$^{-2}$ – which is half the
CL for grain yield loss (LRTAP Convention, 2017) – for at least one year of the century. Then, the ΔPOD$_6$ is calculated as
the difference of each year and the baseline. Finally, the ΔPOD$_6$ timelines of both CMIP6 models are averaged to get the
regional mean trend (thick line). Table 4 lists the mean ΔPOD$_6$ at *2050* (2045-2054) and at *2100* (2090-2099) region by
region. ΔPOD$_6$ follows a decreasing trend under SSP1-2.6 in every region, which is the result of decreasing POD$_6$ down to
near-zero values well before 2100. Trends of ΔPOD$_6$ generally do not decrease under SSP3-7.0, with 9 out of 12 regions
displaying increasing POD$_6$ values at the end of the century. Under SSP5-8.5, ΔPOD$_6$ timelines show a characteristic
downward concave shape (raising and then falling), usually reaching their maxima between 2050 and 2080, before returning
closer to the baseline values (ΔPOD$_6$≈0). This reflects the air quality improvement policies explicitly assumed by this
scenario starting from the mid-of the century (Kriegler et al., 2017).

South Asia shows the greatest POD$_6$ increase across the three scenarios (+0.51 mmol m$^{-2}$). This area reports the maximum
increase at mid-century both under SSP3-7.0 and SSP5-8.5, and the highest absolute increase at the end of the century under
SSP3-7.0. Furthermore, it is one of the two regions with increased POD$_6$ at 2100 for SSP5-8.5, even though not statistically
significant (Table 4). Sub-Saharan Africa is the region with the second highest POD$_6$ increase across SSPs (+0.43 mmol m$^{-2}$
PLA). The final POD$_6$ at the end of the century is 2.9 times the baseline value under SSP3-7.0, the largest relative difference
observed across regions and scenarios. South-East Asia and Central America also emerge as regions with distinctive positive
changes both at mid-century for SSP5-8.5 (+0.58/+0.61), and at the end of the century for SSP3-7.0 (+0.66/+0.45).
Furthermore, South-East Asia is the region with the highest POD$_6$ interannual standard deviation across SSPs (mean SD:
0.27 mmol m$^{-2}$; data not shown), most likely reflecting the largest interannual variability of O$_3$ concentrations and $f_{clim}$
among all regions (mean SD: 1.74 ppb and 0.06, respectively). Noticeably, East Asia is the region with the highest baseline
POD$_6$ value (1.49 ± 0.84 mmol m$^{-2}$). It experiences an increment of +0.41 and +0.52 mmol m$^{-2}$ at mid-century under SSP3-
7.0 and SSP5-8.5 respectively, and a significant decrease of -0.45 mol m$^{-2}$ at 2100 under SSP5-8.5. Central Asia and Middle
East also display a higher POD$_6$ at 2050 and 2100 with respect to the baseline, especially under SSP3-7.0, although only
UKESM1-0-LL shows nodes with POD$_6$>0.65 mmol m$^{-2}$ for at least one year of the century. Arguably, South America and
North Africa are subject to fairly similar POD$_6$ changes compared to the baseline under SSP3-7.0 and SSP5-8.5 (Table 4),
even though only under SSP1-2.6 South America displays statistically significant changes. North America is the only region
with significantly negative POD$_6$ trends in all the considered scenarios, while Russia-Belarus-Ukraine has negative ΔPOD$_6$
under any scenario, even though the change is not significant under SSP3-7.0. On the other hand, it should be noticed that
Europe has non-significant negative changes under SSP3-7.0 and SSP5-8.5, which is the result of opposing trends between
Southern and Northern Europe (see Figure 5).



**Figure 4:** $\Delta POD_6$ **across the century with respect to the baseline average over different SSPs, divided by region; only nodes with** $POD_6 > 0.65$ **mmol m$^{-2}$ in at least one year of the timeline are considered. Bold lines indicate the mean between the results from GFDL-ESM4 and UKESM1-0-LL, thin lines indicate the results from the individual models. In some cases (d,f,i,l), only UKESM1-0-LL had nodes with** $POD_6 > 0.65$ **mmol m$^{-2}$. The number in the upper-left corner of each panel is the mean** $POD_6$ **over the baseline** $\pm$ **spatial SD (mmol m$^{-2}$). These results are from the rain-fed simulation.**



**Table 4: Means ΔPOD$_6$ by region and SSP (mmol m$^{-2}$ PLA) at *2050* (2045-2054) and *2100* (2090-2099) with respect to the *baseline* (2000-2014). The POD$_6$ regional means are calculated only over the nodes with POD$_6$>0.65 mmol m$^{-2}$ PLA during at least one year of the century. These results are from the rain-fed simulation. Starred numbers indicate significant (p<0.05) differences.**

| Region[b] | Baseline POD$_6$ [mmol m$^{-2}$] | 2050 ΔPOD$_6$ [mmol m$^{-2}$] | | | 2100 ΔPOD$_6$ [mmol m$^{-2}$] | | |
|---|---|---|---|---|---|---|---|
| | historical | SSP1 | SSP3 | SSP5 | SSP1 | SSP3 | SSP5 |
| East Asia | 1.49 | -0.64* | +0.41* | +0.52* | -1.22* | +0.21 | -0.45* |
| South-East Asia | 0.96 | -0.16* | +0.61* | +0.58* | -0.59* | +0.66* | -0.20* |
| South Asia | 0.97 | -0.01 | +1.12* | +0.74* | -0.46* | +1.36* | +0.28 |
| Central Asia[a] | 0.55 | -0.25* | +0.65* | +0.27* | -0.38* | +0.58* | -0.06 |
| North America | 1.48 | -1.30* | -0.75* | -0.63* | -1.37* | -0.98* | -0.58* |
| Central America | 0.83 | -0.50* | +0.35* | +0.61* | -0.71* | +0.45* | -0.24* |
| South America[a] | 0.78 | -0.47* | +0.29 | +0.22 | -0.60* | +0.31 | -0.33 |
| Russia-Belarus-Ukraine[a] | 0.92 | -0.71* | -0.15 | -0.27* | -0.81* | -0.22 | -0.30* |
| Europe | 1.07 | -0.89* | -0.20 | -0.13 | -0.97* | -0.25 | -0.05 |
| North Africa | 0.85 | -0.49* | +0.35* | +0.19* | -0.64* | +0.39 | -0.11* |
| Sub-Saharan Africa | 0.49 | +0.06* | +0.35* | +0.78* | -0.24* | +0.93* | +0.57* |
| Middle East[a] | 0.79 | -0.33* | +0.63* | +0.10 | -0.64* | +0.68* | -0.05 |

[a] **Only UKESM1-0-LL has nodes with POD$_6$>0.65 mmol m$^{-2}$ PLA during at least one year of the century.**
[b] **Region definitions are based on those established by the Hemispheric Transport of Air Pollutants (HTAP2; Huang et al., 2017).**

### 3.3 Global POD$_6$ estimates and intermodel comparison

Figure 5 shows POD$_6$ global maps obtained with the O$_3$ stomatal flux model for the baseline period and at the end of the century. During the baseline years, POD$_6$ exceeded 0.65 mmol m$^{-2}$ over the 8.6% and 34.1% of the global wheat area using GFDL-ESM4 and UKESM1-0-LL, respectively (Figure 5a,b). Despite this difference, which is most likely due to the intermodel difference in O$_3$ concentrations, both models identified Eastern North America, Southern Europe and East Asia as hotspots for O$_3$ risk at the beginning of the century. Furthermore, UKESM1-0-LL pinpoints some areas of Sub-Saharan Africa and South America potentially at risk for O$_3$.

Under SSP1-2.6, O$_3$ risk decreases across the whole globe at the end of the century (Figure 5c,d), likely due to declining O$_3$ concentrations (Figure 1c,d), with POD$_6$ exceeding the 0.65 mmol m$^{-2}$ threshold over the 1.8% and the 13.9% of the global wheat area using GFDL-ESM4 and UKESM1-0-LL. Under SSP3-7.0 and SSP5-8.5 (Figure 5e-i), changes are not as straightforward, and they need to be considered region by region, and model by model. The three regions that the baseline



analysis identified as hotspots follow different paths and dynamics in the remaining part of the century. Eastern North

America is the only region with an evident decreasing $O_3$ risk under any scenario, which is due to decreasing $O_3$ concentrations (Figure 1) in a fairly constant $f_{clim}$ (Figure 2). For Eastern China, $POD_6$ either lowers or remains fairly similar to the baseline at the end of the century under both scenarios. The decreasing $POD_6$ is especially evident for SSP5-8.5, comparatively to SSP3-7.0, revealing that the policies controlling $O_3$ precursors emissions (Fujimori et al., 2017) might have a large impact over this region. In Europe, $O_3$ risk hotspots shift from Southern to Northern Europe, following less

limiting temperatures for stomatal $O_3$ uptake at high latitudes (Figure 3c,d). Both GFDL-ESM4 and UKESM1-0-LL recognize an increased risk in the southern and eastern edges of the Tibetan plateau (region from Kashmir to Sichuan) by the end of the century. Within this region, Nepal, Bhutan, and East India reach the $POD_6$ peak value of 8.7 mmol m$^{-2}$ PLA for SSP3-7.0 and 6.3 mmol m$^{-2}$ PLA for SSP5-8.5 at 2100. Furthermore, UKESM1-0-LL identifies areas with increased $O_3$ risk also in Sub-Saharan Africa, Central and South America, especially under scenario SSP3-7.0.

Using the Eq.                           (3, the $POD_6$ in Figure 5 can be converted to relative yield losses (RYL). Based on the $POD_6$ values calculated using UKESM1-0-LL, the highest RYL values (95$^{th}$ percentile) over Europe are estimated about 8.5-9.1%, both at the beginning and at the end of the century under SSP3-7.0 and SSP5-8.5. On the other hand, under SSP1-2.6, the 95$^{th}$ percentile of RYL is substantially lower (1.6%).

Over the entire East Asia, the highest values of RYL are around 16% at the present time, with no relevant decrease for the end of the century under SSP3-7.0 (14.9%). Under SSP5-8.5 and SSP1-2.6 these peak values decrease to 9.8% and 3.9%, respectively, by the end of the century.

Across the southern and eastern edges of the Tibetan plateau, the present-day maximum RYL value is 25.4%, reached in the Sichuan province (China). Over this region, only under SSP1-2.6 the maximum RYL value falls to 18.7% by end of the

century, while it increases to 28.6% and to 31.3% under SSP5-8.5 and SSP3-7.0, respectively.

From the present day to the end of century, the maximum RYL in South America increase from 9.8% to 10.2% under SSP3-7.0, in Sub-Saharan Africa from 8.5% to 12.3% and 15.4% under SSP3-7.0 and SSP5-8.5, and in Central America from 11.2% to 13.8% under SSP3-7.0.

Figure 6 shows the agreement between the two models GFDL-ESM4 and UKESM1-0-LL using the Pearson's correlation

coefficient ($\rho$), calculated in every node for each SSPs. This indicator shows that the two models rarely disagree on the $POD_6$ changes across SSPs, as highlighted by the very few areas with negative correlation coefficients. However, while the correlation for SSP1-2.6 is relatively strong everywhere, for the other scenarios the correlation is high only in East China and in a few spot areas (e.g. Ethiopia $\rho > 0.75$). However, lower correlation coefficients ($0.15 < \rho < 0.45$) are found for Eastern US under SSP3-7.0, and for Europe and South America under SSP5-8.5.





**Figure 5: Mean POD$_6$ over the baseline period (a,b), and ΔPOD$_6$ at 2100 with respect to the baseline across the different SSPs for GFDL-ESM4 (c,e,g) and UKESM1-0-LL (d,f,h). These results are from the rain-fed simulation.**
**Note: Baseline indicates the mean between 2000 and 2014, and 2100 indicates the mean between 2090-2099.**



**Figure 6: Pearson correlation coefficient ($\rho$) between $\Delta POD_6$ at 2100 calculated from GFDL-ESM4 and UKESM1-0-LL. These results are from the rain-fed simulation.**





### 3.4 POD$_6$ under assumption of field capacity

The plant available water in the soil is commonly set to non-limiting conditions in the simplified flux calculations for integrated assessment modelling (POD$_Y$IAM; LRTAP Convention, 2017). This approach estimates POD$_Y$ to provide an

indicative O$_3$ risk assessment at large scale. Due to its assumptions, POD$_Y$IAM indicates the damage under the worst-case scenario, i.e., the potential damage to plants when soil water is not a limiting factor for g$_s$ and O$_3$ uptake. Our 'field capacity' run assumes the same soil water hypothesis as this framework, and the comparison with the results of the rain-fed run offers an estimate of the O$_3$ risk uncertainty related to water availability for plants. In this context, the POD$_6$ under the FC assumption represents the maximum possible O$_3$ risk.

Figure 7 shows the differences between POD$_6$ calculated with the 'field capacity' run, and POD$_6$ calculated with the 'rain-fed' run. Differences between the POD$_6$ calculated through the two runs are shown at each time window – for the baseline and for the end of the century – and for each SSP. Southern Europe (lat.<45°) and South Asia are the regions experiencing the most extensive increase of POD$_6$ in field capacity condition with respect to the rain-fed ones. On average, POD$_6$ under field capacity is $0.74 \pm 0.30$ mmol m$^{-2}$ greater than POD$_6$ under rain-fed conditions in Southern Europe, and $0.76 \pm 0.21$

mmol m$^{-2}$ greater in South Asia, over the whole century, and across SSPs and models. POD$_6$ under field capacity in South-East Asia and in North Africa increases by $1.68 \pm 0.47$ mmol m$^{-2}$, and $1.07 \pm 0.39$ mmol m$^{-2}$. However, only small fractions of these regions are affected, since most of the land-use does not include wheat. Other regions with increased POD$_6$ under the FC assumption are: Middle East ($+0.90 \pm 0.35$ mmol m$^{-2}$), Central America ($+0.90 \pm 0.39$ mmol m$^{-2}$), and Sub-Saharan Africa ($+0.81 \pm 0.19$ mmol m$^{-2}$).

It is important to underline that wheat cultivation under strong water limitations ($f_{soil} < 0.1$; Figure 3i-l) would need additional water sources besides rain. In this case, the 'field capacity' run might be more suitable to assess the potential O$_3$ risk, because POD$_6$ over widely irrigated regions would be underestimated using the rain-fed condition approach. The results in this section (Figure 7) suggest that, regardless of the considered scenario, there would be more potential hotspots worldwide than previously highlighted (Figure 5). For instance, South Asia consistently has the highest O$_3$ concentrations

across the globe (Figure 1), but the rain-fed simulation indicates that g$_s$ is strongly limited by soil water availability, preventing any O$_3$ risk in a region with major wheat production (Monfreda et al., 2008). However, since this region requires extensive irrigation (Brauman et al., 2013; Chiarelli et al., 2020), the POD$_6$ calculated through the field capacity simulation might be more suitable to quantify the O$_3$ risk, indicating that this area might be a further hotspot for O$_3$ damage and food security. Similar conclusions can be drawn for any areas with large differences between field capacity POD$_6$ and rain-fed

POD$_6$, as long as irrigation is supplied for wheat production.



**Figure 7: Mean differences between POD$_{6,FC}$ calculated under FC conditions and the POD$_6$ calculated under rain-fed conditions over the baseline period (a,b), and at 2100 across the different SSPs for GFDL-ESM4 (c,e,g) and UKESM1-0-LL (d,f,h). Baseline indicates the mean between 2000 and 2014, and 2100 indicates the mean between 2090-2099.**



### 3.5 The contribution of radiative forcing and pollutant emissions to POD$_6$

Here we perform an ANOVA (see details in Section 2.5) to estimate which factor among EP, RF and their interaction (I), has the most significant impact on POD$_6$ changes at the end of the century around the globe. The different levels of EP and RF are described in Table 2 for each SSP, and are responsible for the different O$_3$ concentrations and $f_{clim}$ values reported in Figure 1, Figure 2, and Figure A6. Figure 8a shows for each node the factor with the most significant effect (i.e. the lowest p-value among the significant ones) among EP, RF, or their interaction (I), obtained from the two-way ANOVA, while Figure 8b-d display the total variance explained by each factor (R$^2$), allowing further quantification of their contributions to the POD$_6$ changes. The ANOVA highlights different regions where the POD$_6$ changes are mostly driven by the same factor. For example, in East China the decrease of POD$_6$ under SSP1-2.6 and SSP5-8.5 (Figure 4; Figure 5; Table 4) significantly depends on EP. In South America POD$_6$ changes are controlled by EP too. In Eastern U.S. and in Japan the large decrease in the POD$_6$ under any SSP (Figure 5) is controlled by the combined effect of EP and RF (interaction), i.e. the effect of a change in emission policies on the POD$_6$ depends on the given specific climate change signal and viceversa. In Russia-Belarus-Ukraine and in Northern Europe (lat.>45°), changes in POD$_6$ are driven by RF, which largely determines $f_{clim}$ differences across scenarios, with decreases over Russia-Belarus-Ukraine, and increases over Northern Europe. This finding suggests that regions where RF controls POD$_6$ changes could experience an increase in POD$_6$ even under strong EP. This may happen, for instance, not only because of increased $f_{clim}$, but also because O$_3$ increases more as a result of the RF enhancement rather than due to precursors emissions, which would be a climate penalty effect on O$_3$ concentrations (for instance, see Northern Europe under SSP5-8.5, Figure 5h). In regions where POD$_6$ decreases the ANOVA identifies the factors that are mainly responsible for this decrease, such as in East China, Eastern U.S. and Japan. In contrast, where POD$_6$ increases the ANOVA suggests the policies that should be adopted to mitigate O$_3$ risk for crops. For instance, in Northern Europe, where RF is responsible for the POD$_6$ increase, the adoption of stricter reduction policies on greenhouse gases emissions, rather than only the adoption of local EP, could lead to reduction of O$_3$ risk for crops as a climate co-benefit.

However, this ANOVA is performed using the data from UKESM1-0-LL only, because GFDL doesn't include the scenario SSP3-7.0pdSST. Including more models could lead to different results, especially for the climate change component (RF), since it is known that UKESM1-0-LL simulates higher warming compared to many other CMIP6 models (McBride et al., 2021; Scafetta, 2023).



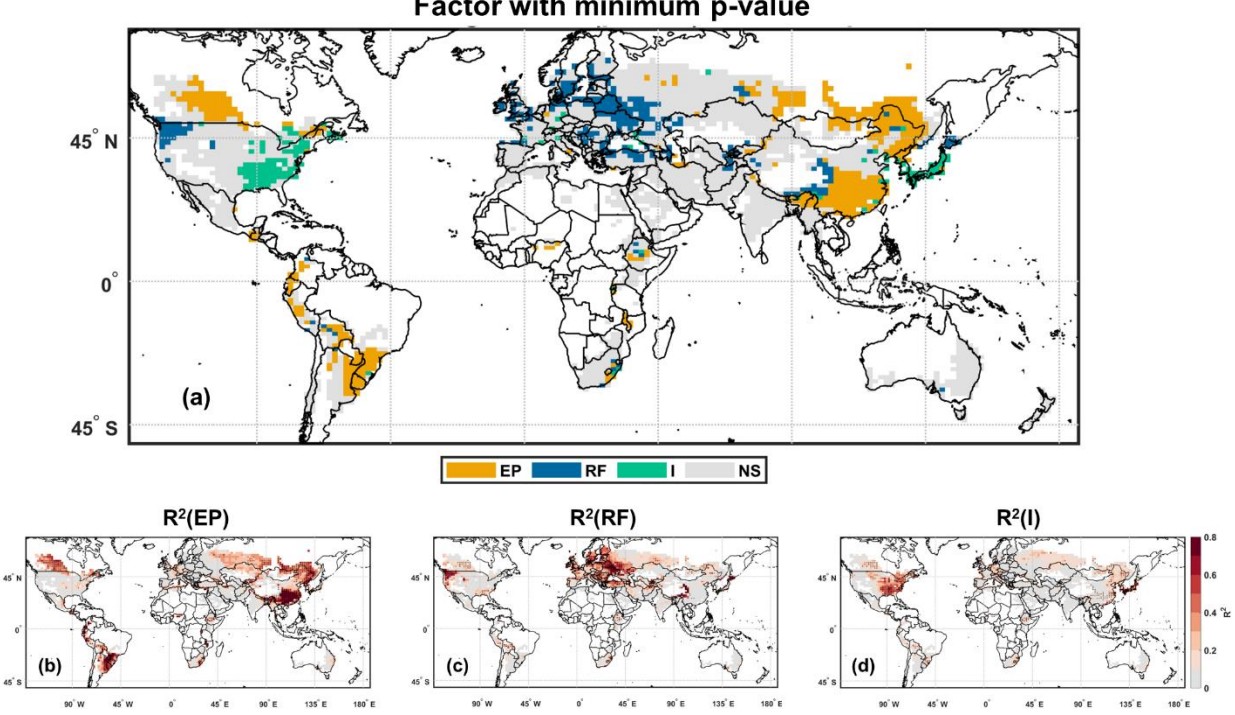

**Figure 8: Map of the factor with the minimum significant p-value (p<0.05, corrected with Bonferroni), determined by the two-way ANOVA (a). Explained variance $R^2$ associated with each factor in each node, with black dots indicating significant p-value (b-d). EP=Emission Policy, RF=Radiative Forcing, I=Interaction, NS=Not Significant.**

## 4. Discussion

This study highlights how wheat in different areas of the globe may be affected by $O_3$ damage in the future. Our results may be associated with different degrees of confidence depending on the agreement between the two available CMIP6 models, and whether an $O_3$ risk is predicted under rain-fed conditions, or just under the FC assumption. Under the rain-fed simulation (and therefore in the FC simulation as well), both the GFDL-ESM4 and UKESM1-0-LL models, consistently identify Europe, East Asia, and the southern and eastern edges of the Tibetan plateau as the main areas at risk throughout the 21st century, despite discrepancies in the estimated $POD_6$ values (Figure 5). The Eastern U.S. is currently among the most critical regions for $O_3$ damage to wheat, but both models project lower $POD_6$ values under all scenarios at the end of the century, suggesting that this area should not be a future concern. Additionally, the FC run indicates that South Asia might be extensively affected by $O_3$ damage in the future. Given the region's widespread need for irrigation (Brauman et al., 2013; Chiarelli et al., 2020), the FC simulation might give a better estimate of $POD_6$ changes over this area, although the $POD_6$ calculated under the FC assumption represents the maximum possible $O_3$ risk and is therefore associated with higher uncertainty. The Indian peninsula and the southern and eastern edges of the Tibetan plateau, being major wheat producers



(Monfreda et al., 2008), should be considered among the areas at highest $O_3$ risk globally across the 21$^{st}$ century. South America, Central America and Sub-Saharan Africa are identified only by UKESM1-0-LL as regions subject to $O_3$ damage to wheat. The increases in POD$_6$, particularly in Central America and Sub-Saharan Africa, are relatively large compared to the baseline and could grow larger under the FC assumption. This fact is particularly concerning for Sub-Saharan Africa, where intense population growth and development is expected during the 21$^{st}$ century (Kc and Lutz, 2017; United Nations Department of Economic and Social Affairs, Population Division, 2022).

Some external sources of uncertainty in this study should also be discussed and are mainly related to the spatiotemporal resolution and the bias of the input variables. The coarser temporal resolution from the CMIP6 models can lead to underestimation of $O_3$ flux peaks and POD$_6$ values (Guaita et al., 2023). Additionally, the coarse spatial resolution of ESMs fails to capture local meteorological features influenced by topology and land cover, and it can misrepresent chemical processes (Brands, 2022). More generally, the coarse resolution is related to biases for any variable, which have been partially evaluated in past literature (e.g. temperature Sellar et al., 2019, O3 in Turnock et al., 2020). Bias correction is beyond the scope of this research; however, we performed a sensitivity analysis by perturbing input variables to evaluate the effect of potential biases on the final POD$_6$ estimates. For the sake of this, we altered separately temperature ($\pm 2°C$), vapor pressure ($\pm 0.2$ kPa), and $O_3$ concentrations ($\pm 5$ ppb) in the rain-fed simulation over the baseline for the UKESM1-0-LL model. The results of these simulations reveal that the standard deviation of the POD$_6$ differences between the perturbed and the original POD$_6$ simulations is smaller than the critical level (1.3 mmol m$^{-2}$ PLA), with temperature the factor contributing the most to the perturbed POD$_6$ response (Figure A5). A more detailed analysis of the sensitivity of POD to ozone and meteorological variables under present-day conditions is being undertaken in a companion study comparing different stomatal ozone flux models [Emmerichs et al., in prep].

Past studies, although using different modelling approaches, generally show good agreement with our results in spatial patterns and identified hotspots of $O_3$ damage (Lombardozzi et al., 2015; Schauberger et al., 2019; Sitch et al., 2007; Tai et al., 2021). More specifically, they identified the eastern U.S., Europe, and East Asia as hotspots, with substantial overlaps. However, there are differences with the available present-day regional assessments that evaluated $O_3$ effects on wheat. For example, POD$_6$ was estimated to reach values up to 8.5 mmol m$^{-2}$ in the Iberian Peninsula (De Andrés et al., 2012), while our results indicate potential threats without reaching such extreme POD$_6$ values. Mills et al. (2011) found that central Europe and Mediterranean coasts are the areas most affected by $O_3$, which is a feature that is only partially represented in our results. On the other hand, our findings are consistent with past estimates of POD$_3$IAM in Sub-Saharan Africa under the field capacity assumption (Sharps et al., 2021b), and with the identified areas most affected by $O_3$ in China (Cao et al., 2024; Qinyi et al., 2023; Wang et al., 2022). The observed differences are likely due to the different spatial resolutions adopted in these regional assessments. Therefore, it is important to emphasize the need of bias-corrected high-resolution spatio-temporal projections of $O_3$ concentrations and meteorology for future regional-scale assessments (Liu et al., 2022).

The most recent literature (Zhou et al., 2024) used ModelE2-YIBs to estimate the $O_3$ effect on GPP, both for the present (2010) and future times (2060) under SSP1-2.6 and SSP5-8.5 scenarios. Their identified spatial patterns are quite similar to



ours, with only minor differences in Western Africa and South America which are likely due to our focus on wheat crops rather than vegetation in general. However, their coarser spatial resolution ($2° \times 2.5°$) might mask some of the spatial patterns identified in our results.

Our results and their consequences for food security can be interpreted in the broader context of the SSPs. Under SSP1 there is a clear decline in $O_3$ risk everywhere, implying a relatively unaffected crop production at the end of the century (globally, 95th percentile of RYL: 2.2%), which would favor food sovereignty, and thus easing adaptation challenges. On the other hand, under SSP3, $POD_6$ would increase throughout the century. This will imply higher crop losses at 2100 for regions with high population growth, particularly South Asia and Sub-Saharan Africa (95th perc. of RYL: 20.0% and 9.3%, respectively).

Since this scenario also includes barriers to trade and to international cooperation, these high food security threats would exacerbate adaptation challenges. With respect to $O_3$ threats, SSP5 is classified as an intermediate scenario, since $O_3$ risk rises to about 2050 in many regions (globally, 95th perc. of RYL: 10.9%) and then reverts to levels similar to or lower than the baseline by the end of the century (7.7%), depending on the area. However, this scenario describes a world that emphasizes technological development, which might also include effective adaptations to food security threats. Ultimately,

SSP3 raises an environmental injustice issue, because it implies that the most affected nations facing food security threats would be the ones with lower resources to address adaptation challenges. On the contrary, a large mitigation effort, such as the one implied within SSP1, would guarantee environmental justice by avoiding food security issues worldwide.

## 5. Conclusions

This study quantified the global and regional evolution of the risk to wheat from $O_3$, expressed as $POD_6$, under different

Shared Socioeconomical Pathways (SSPs) up to 2100. The results show that for SSP1-2.6, the $O_3$ risk will decrease worldwide by the end of the century. Under SSP5-8.5 the $POD_6$ will increase by mid-century before reverting to present-day levels by 2100 in most of the considered regions. On the other hand, under SSP3-7.0, there will be an increase in $POD_6$ over many regions by the end of the century.

In this latter scenario, both CMIP6 models (UKESM1-0-LL and GFDL-ESM4) agree that Northern Europe, East China, and

the southern and eastern edges of the Tibetan Plateau will be the most extensively at risk for future food security, both under rain-fed and field capacity conditions, with $POD_6$ values that will be about six times greater than the current critical level for grain yield loss over the latter region. However, for other regions, like South Asia and Sub-Saharan Africa, the predicted adverse $O_3$ effects on wheat are more uncertain, as they are shown only by one of the CMIP6 models or only occurring under field capacity conditions. Analysis of these regions would benefit from employing finer spatio-temporal resolutions, as

revealed by the comparison made with other regional-scale studies. Finally, although the dry deposition model employed in this research allows explicit evaluation of the $O_3$ risk for wheat, it does not account for any vegetation feedback to $O_3$ concentrations and climate beyond what is already embedded in the CMIP6 models.



More stringent policies to reduce radiative forcing and $O_3$ precursor emissions, as expressed in SSP1-2.6, would lead to almost complete removal of the $O_3$ risk for wheat, thus avoiding $O_3$-induced food security issues worldwide. Conversely, the

control policies directed to limit solely $O_3$ precursors but not greenhouse gases (SSP5-8.5) does not necessarily lead to benefits, as observed for instance in Northern Europe. Finally, the SSP3-7.0 simulations highlight that enhanced radiative forcing and uncontrolled $O_3$ precursor emissions may raise environmental justice issues, as food security threats would exacerbate adaptation challenges for low-income nations.

## 6. Author contributions

PRG and GG conceptualized the study, the methodology, and prepared the software; PRG curated and analysed the data, and created the visualization; PRG and ST performed the investigation and the data collection of UKESM1-0-LL and GFDL-ESM4 outputs; GG administered the project; GG and RM provided the computing resources; GG, RM and PC supervised the research activity; PRG, GG, RM, PC and LZ contributed to the original draft preparation; all authors reviewed and edited the final version of the paper.

## 7. Competing interests

Leiming Zhang is member of the editorial board of journal Atmospheric Chemistry and Physics.

## 8. Code and Data availability

The data for individual years used to produce the figures in this study is available at https://doi.org/10.5281/zenodo.13485000. The Matlab code for the model can be made available upon request to the authors.

## 9. Acknowledgements

We thank dr. Shengchao Qiao and colleagues for their valuable contribution in providing the sowing dates dataset of wheat and for their useful comments to adapt their results to our case-study. This manuscript benefits from discussions during the meeting of the Ozone Deposition Focus working group of the Tropospheric Ozone Assessment Report, phase II. We would especially like to thank prof. Lisa Emberson for her fruitful insights on this work. We are grateful to Cazimir Kowalski for

offering helpful feedback on the writing style and structure of this manuscript.

This work was also supported by the "5x1000 funding to research" of the Italian Government and by Catholic University of the Sacred Heart in the frame of its Programs of promotion and dissemination of the scientific research (Funding line D3.1).





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





## Appendix A

**Figure A1:** $f_{temp}$ **mean values over the baseline period (a,b), and** $\Delta f_{temp}$ **at 2100 with respect to the baseline across the different**
**SSPs for GFDL-ESM4 (c,e,g) and UKESM1-0-LL (d,f,h).** $f_{temp}$ **ranges from 0 to 1 depending on whether it is limiting or not.**
**From this study's perspective, positive or negative** $\Delta f_{temp}$ **correspond to higher and lower O3 risk respectively (red and blue, in c-**
**h).**
**Note: Baseline indicates the mean between 2000 and 2014, and 2100 indicates the mean between 2090-2099.**



**Figure A2:** $f_{VPD}$ **mean values over the baseline period (a,b), and** $\Delta f_{VPD}$ **at 2100 with respect to the baseline across the different SSPs for GFDL-ESM4 (c,e,g) and UKESM1-0-LL (d,f,h).** $f_{VPD}$ **ranges from 0 to 1 depending on whether it is limiting or not. From this study's perspective, positive or negative** $\Delta f_{VPD}$ **correspond to higher and lower O3 risk respectively (red and blue, in c-h). Note: Baseline indicates the mean between 2000 and 2014, and 2100 indicates the mean between 2090-2099.**



**Figure A3:** $f_{soil}$ **mean values over the baseline period (a,b), and** $\Delta f_{soil}$ **at 2100 with respect to the baseline across the different SSPs for GFDL-ESM4 (c,e,g) and UKESM1-0-LL (d,f,h).** $f_{soil}$ **ranges from 0 to 1 depending on whether it is limiting or not. From this study's perspective, positive or negative** $\Delta f_{soil}$ **correspond to higher and lower O3 risk respectively (red and blue, in c-h). Note: Baseline indicates the mean between 2000 and 2014, and 2100 indicates the mean between 2090-2099.**







**Figure A4: Mean POD$_0$ over the baseline period (a,b), and ΔPOD$_0$ at 2100 with respect to the baseline across the different SSPs for GFDL-ESM4 (c,e,g) and UKESM1-0-LL (d,f,h). These results are from the rain-fed simulation.**
**Note: Baseline indicates the mean between 2000 and 2014, and 2100 indicates the mean between 2090-2099.**



**Figure A5: Standard Deviation (SD) of yearly differences of POD₆ over the baseline between all the perturbed simulations and the rain-fed simulations (a). Mean differences of POD₆ over the baseline between the perturbed simulations and the rain-fed simulations for each variable (b-g).**
**Note: Baseline indicates the mean between 2000 and 2014.**



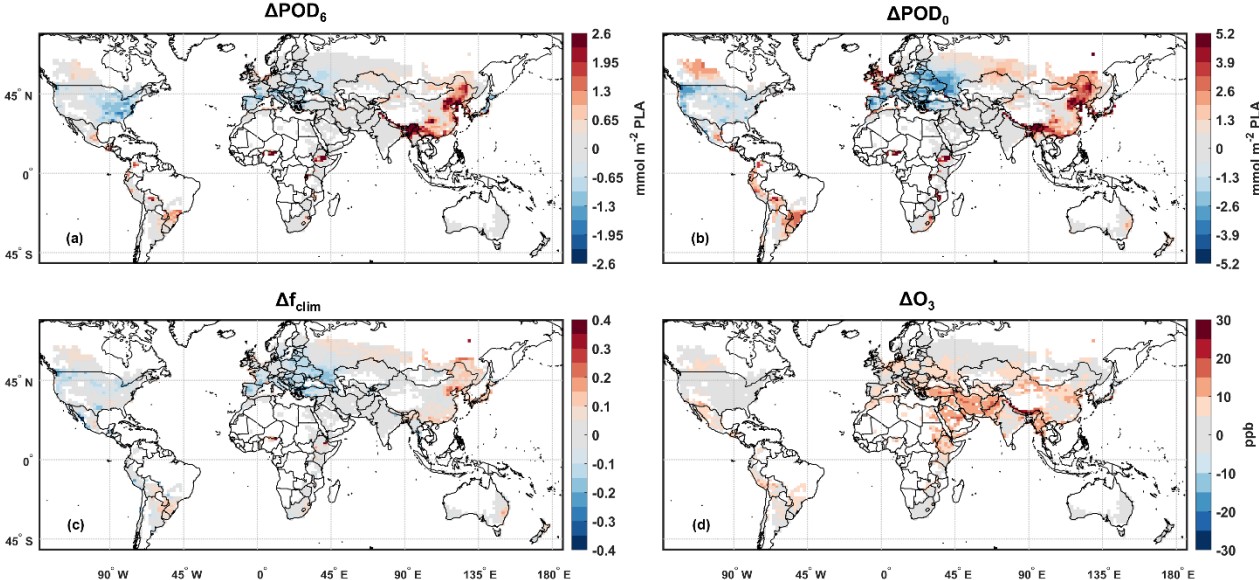

**Figure A6: values at 2100 for the experiment SSP3-7.0pdSST with respect to the baseline of (a) ΔPOD$_6$, (b) ΔPOD$_0$, (c) Δf$_{clim}$, (d)**
**ΔO$_3$. These results are from the rain-fed simulation.**

**Appendix B**

The conversion from model levels to geometric height above the ground is achieved using the hydrostatic assumption and
integrating the following equation from the surface pressure to the actual pressure at the model level:

$$dz = \frac{R_d T_v(p)}{gp} dp \qquad\qquad\qquad (B1)$$

Here, $p$ denotes pressure, $R_d = 287$ J kg$^{-1}$K$^{-1}$ is the specific gas constant of dry air, $g = 9.81$ m s$^{-2}$ is the acceleration due
to gravity and it is assumed constant since calculations are performed for the model levels closest to the ground, and thus
there is no need to incorporate geopotential height. $T_v(p)$ [K] denotes the virtual temperature profile, obtained by
interpolating between model levels after converting the values from air temperature $T$ [K] through $T_v = T(1 + 0.609133q)$,
with $q$ being the specific humidity [kg H$_2$O kg$^{-1}$ air].

UKESM1-0-LL explicitly indicates that the near-surface model level for O$_3$ concentration is at 20 m above the ground, and
therefore no calculation for geometric height is required.