# Peer review of "Global flux-based ozone risk assessment for wheat up to 2100 under different climate scenarios"

_EGUsphere, 2024_

## Community Comment (CC1)

October 28, 2024

Comments by Owen R. Cooper (TOAR Scientific Coordinator of the Community Special Issue) on:

**Global flux-based ozone risk assessment for wheat up to 2100 under different climate scenarios**

Pierluigi Renan Guaita, Riccardo Marzuoli, Leiming Zhang, Steven Turnock, Gerbrand Koren, Oliver Wild, Paola Crippa, and Giacomo Alessandro Gerosa

EGUsphere [preprint], https://doi.org/10.5194/egusphere-2024-2573
Discussion started Aug. 30, 2024
Discussion closes Oct. 31, 2024

This review is by Owen Cooper, TOAR Scientific Coordinator of the TOAR-II Community Special Issue. I, or a member of the TOAR-II Steering Committee, will post comments on all papers submitted to the TOAR-II Community Special Issue, which is an inter-journal special issue accommodating submissions to six Copernicus journals:  ACP (lead journal), AMT, GMD, ESSD, ASCMO and BG. The primary purpose of these reviews is to identify any discrepancies across the TOAR-II submissions, and to allow the author teams time to address the discrepancies.  Additional comments may be included with the reviews. While O. Cooper and members of the TOAR Steering Committee may post open comments on papers submitted to the TOAR-II Community Special Issue, they are not involved with the decision to accept or reject a paper for publication, which is entirely handled by the journal's editorial team.

**Comments regarding TOAR-II guidelines:**

TOAR-II has produced two guidance documents to help authors develop their manuscripts so that results can be consistently compared across the wide range of studies that will be written for the TOAR-II Community Special Issue.  Both guidance documents can be found on the TOAR-II webpage: https://igacproject.org/activities/TOAR/TOAR-II

*The TOAR-II Community Special Issue Guidelines*:   In the spirit of collaboration and to allow TOAR-II findings to be directly comparable across publications, the TOAR-II Steering Committee has issued this set of guidelines regarding style, units, plotting scales, regional and tropospheric column comparisons, and tropopause definitions.

*The TOAR-II Recommendations for Statistical Analyses*:  The aim of this guidance note is to provide recommendations on best statistical practices and to ensure consistent communication of statistical analysis and associated uncertainty across TOAR publications. The scope includes approaches for reporting trends, a discussion of strengths and weaknesses of commonly used techniques, and calibrated language for the communication of uncertainty. Table 3 of the TOAR-II statistical guidelines provides calibrated language for describing trends and uncertainty, similar to the approach of IPCC, which allows trends to be discussed without having to use the problematic expression, "statistically significant".

**General comments:**

At the time of this writing the authors have already received two very thorough reviews from the anonymous referees. Therefore, I will limit my comments to the two topics described below.

1) As stated in the first paragraph of *The TOAR-II Community Special Issue Guidelines*:
"as TOAR papers and the Copernicus journals focus on science and not policy, the submitted paper may be policy-relevant, but not policy-prescriptive."
In light of this guideline, the final sentence of the abstract should be re-written as it could be interpreted as a policy recommendation:
"Therefore, this study emphasizes the need for effective emission mitigation policies of both $O_3$ precursors and greenhouse gases to preserve global food security from $O_3$ damages."
For example, the following suggestion shows how the findings from the analysis can be described as policy-relevant, without making any policy prescriptive statements:
"These findings are relevant to policymakers as they indicate the potential impacts of air pollution and climate change on crop productivity and food security."

2) The following discussion is aimed at reporting trends according to the TOAR-II guidelines. While the submission by Guaita et al. (2024) does not specifically report trends (e.g. change in ozone per decade), ANOVA is used in a similar way to quantify the differences between present-day and future ozone projections.
The expression "statistically significant" is used throughout the submitted manuscript, however this expression is now recognized as being problematic and it should be abandoned and replaced by the more helpful method of reporting all trends (or ANOVA results), along with $p$-values and uncertainty estimates (e.g. 95% confidence intervals), followed by a discussion of the trends and the author's opinion regarding their confidence in the trend values. This advice comes from a highly influential paper by Wasserstein et al. (2019), published in the journal, *The American Statistician*, that has already been cited over 1900 times (according to Web of Science). This advice was adopted by the first phase of TOAR (Tarasick et al., 2019) and is also being used by TOAR-II. Some other recent papers on ozone trends that have taken this advice are: Chang et al., 2020; Cooper et al., 2020; Gaudel et al., 2020; Chang et al., 2022; Wang et al., 2022; Chang et al., 2024; Seguel et al., 2024. Because these papers report all trend values, uncertainties, and all $p$-values, and also discuss the trend results, there is no confusion regarding the findings, and one does not even notice that the term "statistically significant" is not used at all. Table 3 of the TOAR-II statistical guidelines provides calibrated language for describing trends and uncertainty, similar to the approach of IPCC.

**References**
Chang, K.-L., et al. (2020), Statistical regularization for trend detection: An integrated approach for detecting long-term trends from sparse tropospheric ozone profiles, Atmos. Chem. Phys., 20, 9915–9938, https://doi.org/10.5194/acp-20-9915-2020
Chang, K.-L., O. R. Cooper, A. Gaudel, M. Allaart, G. Ancellet, H. Clark, S. Godin-Beekmann, T. Leblanc, R. Van Malderen, P. Nédélec, I. Petropavlovskikh, W. Steinbrecht, R. Stübi, D. W. Tarasick, C. Torres (2022), Impact of the COVID-19 economic downturn on tropospheric ozone trends: an uncertainty weighted data synthesis for quantifying regional anomalies above western North America and Europe, *AGU Advances, 3*, e2021AV000542. https://doi.org/10.1029/2021AV000542

Chang, K.-L., Cooper, O. R., Gaudel, A., Petropavlovskikh, I., Effertz, P., Morris, G., and McDonald, B. C.: Technical note: Challenges in detecting free tropospheric ozone trends in a sparsely sampled environment, Atmos. Chem. Phys., 24, 6197–6218, https://doi.org/10.5194/acp-24-6197-2024, 2024.

Cooper, et al. 2020. Multi-decadal surface ozone trends at globally distributed remote locations. Elem Sci Anth, 8: 23. DOI: https://doi.org/10.1525/elementa.420

Gaudel, A., et al. (2020), Aircraft observations since the 1990s reveal increases of tropospheric ozone at multiple locations across the Northern Hemisphere. Sci. Adv. 6, eaba8272, DOI: 10.1126/sciadv.aba8272

Seguel, R. J., Castillo, L., Opazo, C., Rojas, N. Y., Nogueira, T., Cazorla, M., Gavidia-Calderón, M., Gallardo, L., Garreaud, R., Carrasco-Escaff, T., and Elshorbany, Y.: Changes in South American surface ozone trends: exploring the influences of precursors and extreme events, Atmos. Chem. Phys., 24, 8225–8242, https://doi.org/10.5194/acp-24-8225-2024, 2024.

Tarasick, D. W., I. E. Galbally, O. R. Cooper, M. G. Schultz, G. Ancellet, T. Leblanc, T. J. Wallington, J. Ziemke, X. Liu, M. Steinbacher, J. Staehelin, C. Vigouroux, J. W. Hannigan, O. García, G. Foret, P. Zanis, E. Weatherhead, I. Petropavlovskikh, H. Worden, M. Osman, J. Liu, K.-L. Chang, A. Gaudel, M. Lin, M. Granados-Muñoz, A. M. Thompson, S. J. Oltmans, J. Cuesta, G. Dufour, V. Thouret, B. Hassler, T. Trickl and J. L. Neu (2019), Tropospheric Ozone Assessment Report: Tropospheric ozone from 1877 to 2016, observed levels, trends and uncertainties. Elem Sci Anth, 7(1), DOI: http://doi.org/10.1525/elementa.376

Wang, H., Lu, X., Jacob, D. J., Cooper, O. R., Chang, K.-L., Li, K., Gao, M., Liu, Y., Sheng, B., Wu, K., Wu, T., Zhang, J., Sauvage, B., Nédélec, P., Blot, R., and Fan, S. (2022), Global tropospheric ozone trends, attributions, and radiative impacts in 1995–2017: an integrated analysis using aircraft (IAGOS) observations, ozonesonde, and multi-decadal chemical model simulations, Atmos. Chem. Phys., 22, 13753–13782, https://doi.org/10.5194/acp-22-13753-2022

Wasserstein, R. L., Schirm, A. L., and Lazar, N. A.: Moving to a world beyond p < 0:05, Am. Stat., 73, 1–29, https://doi.org/10.1080/00031305.2019.1583913, 2019.

---

## Author Comment (AC2)

**General comments**

This manuscript describes a risk assessment exercise of ozone effects on wheat production at the global scale based on modelled phytotoxic ozone dose (POD6). POD6 values are calculated following the CLRTAP methodology, using as input data the ozone concentration and other meteorological variables produced by two Earth system models, run under different shared socioeconomic pathways with contrasting radiative forcing and atmospheric pollution control policies over the XXI century. Changes in POD6 under the different scenarios are described for different regions of the world. The influence of meteorological conditions and ozone concentration on the final POD6 estimate and relative wheat yield loss are also analysed. This kind of analysis is useful to understand how climatic changes may affect the negative effects of tropospheric ozone pollution on agricultural yield in the future rather than focusing solely on changes in ozone concentration. The analysis provides also insights on the relative weight of air pollution policies and radiative forcing scenarios on the risk of ozone effects on agricultural yields by the end of the XXI century. The relative influence of these two factors changes for different regions of the world, highlighting the co-benefits of controlling both ozone precursor and greenhouse gases emissions to the atmosphere.

The manuscript is well structured and written and make a relevant contribution to the field of risk assessment effects on agricultural production under future climatic conditions. The objectives, datasets, methods and results are clearly presented and easy to follow, although I have some comments that may need minor revisions of the manuscript:

We thank the reviewer for their positive feedback and helpful comments which we address below.

**Specific comments**

Line 138 and Table 1. From the list of variables in Table 1, I am not able to find soil moisture. Thus, I understand that soil moisture was modelled based on GFDL and UKESM outputs. Please explain, or cite methods, on how soil moisture was modelled and how plant available water was computed from soil moisture data.

The water in the soil that is available to the plant was calculated as a water budget through a simple bucket model (following the approach by Mintz and Walker, 1993). We deemed this method appropriate, since we are simulating a wheat field, whereas the soil moisture output from ESMs takes into account many different land-use types within the same tile.

An additional explanation was added to the manuscript in Section 2.1.

**Line 169. Crop geometry refers to the development of LAI and SAI over the course of the growing season, as a function of thermal time, as described in Guaita et al 2023? If that is the case, I would specify this somewhere in the manuscript, since the term "crop geometry" seems confusing to me. Was plant height changed as well over the course of the growing season?**

Thanks for pointing that out. Yes, Guaita et al. (2023) is the reference also for the crop geometry. This includes LAI, SAI, root depth and plant height. The term "crop geometry" was removed from the manuscript and specification of the LAI, SAI, root depth and plant height were added in the sentence of Section 2.3.

**Line 176. The Mediterranean wheat parameterization for soil moisture refers to volumetric water content while in this modelling exercise, plant available water is used. Please describe how was this modified in your calculations.**

The reviewer is correct. For the mediterranean wheat, our model calculates SWC as the soil water available to the plant (i.e. the difference between the actual SWC and the SWC at wilting point) instead of PAW. This is effectively a difference compared to Guaita et al. (2023), for which only the continental parameterization of winter wheat was adopted. It is now specified at the end of the section 2.3, in the last paragraph about model updates.

For clarity, we replaced the wording "plant available water in the soil" with the more general "soil water available to the plant" across the whole manuscript.

**Line 191. It seems that there is a typo in the reference Guaita et al 2023b, or the reference is missing in the bibliography.**

Thanks. Exactly, that's a typo, and it has been corrected.

**Line 215. I think this sentence needs some clarification. The prescribed sowing dates come from Qiao et al. (2023), while thermal times describing the phenological development of wheat were taken from González-Fernández et al., 2013 and Grünhage et al., 2012). Thus, I understand that if the thermal time at leaf senescence (according to González-Fernández et al., 2013 and Grünhage et al., 2012) was not reached before the following prescribed sowing date (according to Qiao et al 2023), then the node was excluded from POD6 calculation.**

Yes, it is exactly as the reviewer said, thank you. The mentioned sentence of section 2.4 has been adjusted accordingly to make it clearer.

**Line 221. Unfortunately, I am not familiar with this sort of spatial analysis to assess its use, although the concept and results obtained look reasonable to me. However, I miss some comments about the tests conducted to assess the assumptions of the ANOVA analysis.**

The data in every node was tested for normality with either the Shapiro-Wilk or the Shapiro-Francia test, depending on the data being either platykurtic or leptokurtic.  In case the test failed, the data in that node was log-transformed. A couple of sentences were added to the manuscript in the methodology section dealing with ANOVA (2.5) to clarify this process.

**Line 239. How the accumulation period changed between the baseline and the 2100 scenarios? Was it earlier and shorter, or longer in different regions of the world?**

The accumulation period shortened up to 5 days on average by the end of the century. At the same time, the accumulation period onset was noticeably earlier, up to 26 days earlier at the end of the century under SSP3-7.0 for UKESM1-0-LL.

We added some sentences on this matter at the end of Section 3.1 and, accordingly, we added more rows Table 3 displaying the duration of the accumulation period at the baseline and at the end of the century for both models. The differences in the number of days between the onset of the accumulation period at the end of the century and at the baseline were also added to the table.

**Lines 244-247. This statement might be more appropriate for the discussion section.**

The reviewer is correct and we appreciate the suggestion. The statement has been moved to the discussion section. Since the discussion section appeared a more appropriate part to elaborate further on the biases, we added a comment regarding the ESMs performances in simulating mean ozone concentrations.

**Line 272. Desert regions or arid regions? In desert regions, it will be unlikely to find rainfed wheat crops?**

Thanks, arid is more appropriate. Clearly, rainfed wheat crops cannot be find in desert regions, the reviewer is correct.

**Figure 1. Please explain in caption the meaning of areas in white.**

Thanks. The explanation has been added.

**Table 4. Means presented in Table 4 are averaged between the two CMIP6 models? Please clarify this.**

Yes, they are averaged between models. The specification has been added to the table.

**Lines 366-369. The mean or median relative yield loss could be also a very helpful metric to describe the range of the expected risk to wheat production**

It is true that it could be useful, and, when possible, the mean RYL have been indicated in the text across the whole section.

**Line 379. How should the very low Pearson correlation coefficients 0.15 > r > -0.15 covering relatively large areas in the map on Figure 6 be interpreted? Is this also reflecting a lack of agreement between models? Higher uncertainty in the expected changes?**

While high Pearson correlation coefficients indicate that the two models are strongly correlated (positively or negatively), very low values indicate that the data is (linearly) uncorrelated. However, this fact can be associated with one of three cases: (1) on average, there are small $\Delta POD6$ values (<0.65 mmol m$^{-2}$) in both models; (2) on average, there are large $\Delta POD6$ values in both models (but linearly uncorrelated among them); (3) there are large $\Delta POD6$ values in only one of the two models.

In the first case, correlation is small because there are small predicted POD6 changes. However, this does not indicate that there is no agreement between models, but rather that both models agree that the change is small.

In the second case, there is an interannual disagreement in the $\Delta POD6$ values between the two models, indicating that there might be disagreement on a year-by-year basis. In this sense, both models might agree on magnitude the expected value, although there is high uncertainty on individual years.

In the third case, there is an actual disagreement that is not highlighted by a negative correlation coefficient: in fact, one of the models predicts a large $\Delta POD6$ on average, while the other one expects values that are close to zero.

However, in our study, 68.6% of the nodes with a low Pearson correlation coefficient pertain to the first case, i.e. both models agree that the POD6 changes are small. 8.4% nodes fall within the second case, and 99.9% of the times the average POD6 changes have the same sign. The remaining 23.0% of the times the two models disagree, in the sense that one of the two expects small POD6 changes, while the other one expects large POD6 changes.

An additional explanation has been added at the end of section 3.3 to clarify this matter.

**Lines 403-406. The description of POD6 increases under non-limiting soil moisture conditions is confusing to me. Most extensive increases in POD6 happens in Southern Europe and South Asia, but the average value of increase seems lower than for other regions like South-East Asia or North-Africa. The values reported are averaged also across models?**

With the term "extensive", we refer to the fraction of the region (with respect to the overall region surface) affected by POD6 increases under FC condition. In order to avoid confusion, we replace the word "extensive" with "widespread", and specified the fraction of the region affected by POD6 increases. The values reported are averaged across models, and it has been specified so in the text.

**Line 447. Does not**

Corrected, thank you.

**Line 455. The discussion should include some comments regarding the performance of the models chosen in this study to simulate ozone concentrations (as mentioned in line 245).**

Thanks for the suggestion. The part relative to the model performances in simulating ozone concentrations has been expanded.

**Also, uncertainties stemming from the use of particular models: in this study the general pattern matches, but one model predicts bigger changes in POD6 and higher risks of O3 effects on yield compared with the other one, and large areas show small Pearson's correlation coefficients between modes, as shown in figure 6. I wonder if multi-model ensembles could be a useful tool for future projections.**

It is a matter of fact that the UKESM1-0-LL predicts generally higher O3 concentrations with respect to GFDL-ESM4, and therefore the POD6 estimated with the first model is higher than the second one. This feature was made explicit in section 4 (discussion). Furthermore, following the second reviewer's request, we performed a new model evaluation which can be found in the supplementary materials, and that will provide further context to interpret discrepancies across models.

The variability between the models is a source of uncertainty for the risk assessment. Hence, we agree with the reviewer that a multi-model ensemble would be welcome for this risk assessment. However, it should be noted that to date only two coupled-chemistry models provide in output hourly O3 concentrations up to 2100, and they are UKESM1-0-LL and GFDL-ESM4.

Regarding the disagreement indicated by the Pearson correlation coefficients between models, there are relatively small fractions of the globe (between 4.3% and 14.3%, depending

on the scenario considered) where the two models disagree, and they can be identified either from negative Pearson correlation coefficients (Figure 6), or from small $POD_6$ changes predicted by one model, and large by the other. However, the position of these nodes did not reveal any specific spatial patterns, suggesting that at least a fraction of the disagreement between the models might be due more to randomness rather than by actual different ESMs features.

Therefore, despite the two models predict different absolute POD6 values, their spatial patterns well agree across the scenarios in the main risk areas.

These features were added to the discussion section.

**Also, there is one variable that will change depending on SSP scenarios compared with the current situation that also affects stomatal conductance and likely PO6 but is not taken into account with this methodology, such as the CO2 concentration. This should be considered an additional uncertainty and limitation of this approach.**

The reviewer is correct. The caveat has been added to the discussion section.

**On the positive side, it would be interesting to comment on the advantages of assessing tropospheric ozone effects on agricultural yield under future climatic conditions using the POD approach as compared with other assessments based only on changes in ozone concentration. Finally, it could be stressed that the results presented here support the co-benefits of abating greenhouse gases and air pollution emissions jointly to help in the mitigation of air pollution effects in agriculture.**

The reviewer is correct and we appreciate the suggestion. We believe that the advantage of using the POD approach resides in the ability to understand that the mitigation of ozone damage also comes as a co-benefit in controlling GHG emissions, at least in some regions of the globe. This has been added to the manuscript.

**Line 479. Subscript missing in O3.**

Fixed, thank you.

---

## Author Comment (AC3)

**This paper presents calculations of POD6 from current conditions to the year 2100. Although changes in crop yield due to ozone over this period are of interest, I find it difficult to know what to make of the results when I have not been given any impression about whether the models can actually predict ozone and especially POD6 to any satisfactory degree in the base run. The lack of comparison with measurements is even more surprising given that this manuscript was submitted to the TOAR-II Special issue! I am afraid I find this omission to be too significant to be ignored, and therefore cannot recommend this manuscript for publication. More detailed comments follow.**

Here we clarify key features of our modeling choices and tools used and summarize key results of the new analyses performed to address the issues raised. Detailed answers can be found in the point-by-point responses and in the newly written Supplementary materials.

The skill of CMIP6 models (UKESM1-0-LL and GFDL-ESM4) in predicting ozone concentration during the baseline period has been evaluated against TOAR ozone data in Turnock et al. (2020) study, which we reference. That study underwent peer review and was published in Atmospheric Chemistry and Physics, so we believe it provides reliable results that we confidently reference.

Therefore, our work acknowledges and builds on those findings and assumes those bias estimates as valid, with the noted uncertainties. Namely, Turnock et al. (2020) reported that CMIP6 models generally overestimate $O_3$ concentrations, although their comparison is performed between the $O_3$ concentrations at the lowest model level (roughly between 15 and 20 m above the ground), and the measured $O_3$ concentrations in the TOAR database (typically between 2 and 3 meters). Since $O_3$ concentrations are lower the closer to the ground, and since our dry deposition model scales $O_3$ concentrations from the lowest model level to the canopy height, this overestimation would reasonably be mitigated. Furthermore, it should be noticed that, despite the acknowledged uncertainties, the ozone data used in this study represent the only currently available coupled-chemistry datasets offering both hourly resolution and the required time span.

Although we believe that further evaluation is unnecessary for the outlined reasons, we provide an additional assessment of the bias between the ozone concentrations predicted by CMIP6 models (scaled from lowest model level to the canopy height) and those measured at stations in the TOAR-II database (scaled to the canopy height as well), according to the reviewer's request. This comparison shows that, as expected, the $O_3$ concentrations modelled by the CMIP6 models have a smaller bias (global averages: +1.31 and +0.88 ppb in GFDL-ESM4 and UKESM1-0-LL, respectively; see supplementary materials attached at the end of this document for details), compared to the overestimation reported by Turnock et al. (2020). Furthermore, we estimate the effect of the $O_3$ bias on the $POD_6$ estimate, and show that it is relatively small on average, due to climate limiting stomatal conductance.

While regional differences in $O_3$ concentrations and associated bias against observations are present, East Asia shows the largest bias, but, in fact, the propagated bias leads to relatively

low overestimation of $POD_6$ (mean: +0.28 and +0.33 mmol m$^{-2}$, for GFDL-ESM4 and UKESM1-0-LL respectively). Therefore, the biases found in the modelled $O_3$ concentrations are acceptable for the purposes of of our study. The main findings regarding the bias will be mentioned in the manuscript, both in the results and in the discussion section, and the supplementary materials will be referenced therein.

Regarding the $POD_6$ metric used to compute the $O_3$ damage, we clarify that it is calculated based on a modified version of the DO3SE model and following the same formulation (as stated in the manuscript). As this model is widely accepted as reliable for predicting ozone deposition against field measurements (as affirmed by the reviewer and the literature), results in our study will be similarly valid as based on a fundamentally analogous deposition model.

Nevertheless, we provide additional evidence on the validity of our dry deposition model by assessing its performances against data from a measurement campaign over a wheat field (Gerosa et al., 2003) Specifically, we compare modelled and observed values of total $O_3$ flux, latent heat flux and friction velocity on a half-hour basis, as detailed in the Supplementary Material and responses below. These variables were chosen because they are directly measured, while on the contrary stomatal conductance and stomatal ozone fluxes are derived from water and $O_3$ fluxes under some assumptions using an indirect inferential methodology. Our results indicate high skills in reproducing the latent heat flux ($R^2$=0.74, and MAE =0.02 W/m$^2$; see supplementary materials attached at the end of this document for details), which is a key proxy for the stomatal conductance, and thus of stomatal fluxes. Therefore, this further supports the appropriateness of using our model. These results are comparable with the ones reported by Mills et al. (2018, with $R^2$=0.63, also cited by the reviewer in the major comments). This similarity was expected, since Mills et al. (2018) employed the DO3SE model, of which our dry deposition model is a modified version.

Moreover, a companion TOAR-II paper will evaluate specifically the performances and the sensitivity to input variables of different stomatal models, including the DO3SE (Emmerichs et al., in prep). We remark that the main objectives of this work are to evaluate the POD for bread wheat across the 21$^{st}$ century, and to identify the regions vulnerable to future food security threats, and that the evaluation of the sensitivity to input variables, and an in-depth evaluation of the performances of the CMIP6 models are beyond the scope of this work.

**Major comments**

**1. The major weaknesses of this paper are that the authors have chosen to model a very difficult ozone metric (POD6), and they present no evidence to show that the models used have any ability to model that metric (or indeed any other), even for present day conditions.**

The POD6 metric is indeed more difficult to calculate, especially in comparison to exposure-based metrics. However, flux-based metrics are frequently referred to as the correct approach to study $O_3$ risk to vegetation (Emberson, 2020; Mills et al., 2011). As mentioned above, our model follows the same paradigm of the DO3SE model, which is widely employed in ozone risk assessments to vegetation, and constitutes the $O_3$ dry deposition scheme within of the EMEP chemical transport model (Simpson et al., 2012), which is cited as a positive example by the reviewer in the Major comment #4. An evaluation of the dry deposition model is now presented in the supplementary materials, which are also attached at the end of the response.

**2. First, about the metric itself. Why was POD6 chosen? It is well known that ozone metrics such as PODY can be very difficult to estimate, especially when the Y threshold is very high (e.g. Sofiev and Tuovinen et al, 2001, Tuovinen 2000, Touvinen et al, 2007). POD is also a difficult metric to obtain from observations because its calculation requires a large number of parameters, assumptions and auxiliary measurements that are usually not available. Such problems explain why the otherwise comprehensive TOAR database of vegetation-relevant ozone metrics (Lefohn et al, 2018, Mills et al, 2018a) did not include estimates of POD.**

The reviewer seems to refer to TOAR-I. In the current TOAR version (TOAR-II), POD is fully recognized as the only metric effectively bridging atmospheric chemistry with plant physiology (e.g., Emberson, 2020; Mills et al., 2011). The harmful effects of ozone on vegetation are not due to the mere exposure to high ozone concentrations, but to the ozone uptake through stomatal pores. The concept of dose, therefore, augment the one of exposure (reflected in metrics like M7, AOT40, etc.), as documented by the ICP Vegetation (LRTAP Convention, 2017) and extensively reported in related papers. This distinction is explicitly noted in the mapping manual, which is cited by the reviewer: "Scientific evidence suggests that observed effects of $O_3$ on vegetation are more strongly related to the uptake of $O_3$ through the stomatal leaf pores (stomatal flux) than to the concentration in the atmosphere around the plants (Mills et al., 2011b)." (see chapter II.3.1.2 of the Mapping manual, whose insert is reported below). This is the main reason we chose to use POD, specifically POD6, as recommended by the mapping manual for wheat.

**III.3.1.2  METRICS FOR CRITICAL LEVELS OF $O_3$ FOR VEGETATION**

A glossary for all terms used for $O_3$ critical levels is provided in Annex III.2.

For $O_3$, two types of metrics are available for risk assessment, either based on the cumulative stomatal flux or the cumulative exposure. Scientific evidence suggests that observed effects of $O_3$ on vegetation **are more strongly related to the uptake of $O_3$ through the stomatal leaf pores (stomatal flux) than to the concentration** in the atmosphere around the plants (Mills et al., 2011b). Stomata are physiologically controlled

**3. Further, the LRTAP mapping manual makes it clear that the so-called PODYSPEC metrics (including POD6SPEC) are intended for situations where ozone and meteorological variables can be accurately estimated at the flag leaf of a wheat plant.**

Ozone and meteorological variables are indeed estimated and referenced at canopy height and flag leaves for wheat fields. This was accounted for by applying a resistive network and the big-leaf approach, as done in DO3SE and also described in detail in our previous paper (Guaita et al., 2023).

**Global scale model simulations are not at all well suited to making accurate predictions of POD6. Indeed, the LRTAP manual suggests that large-scale simulations make use of the a lower Y threshold, and some simpler parameter settings, which they denoted POD3IAM.**

We disagree, as the LRTAP manual explicitly advocates for using our approach. Specifically, text Box 9 on page 45 (reported below, LRTAP Convention, 2017) explicitly recommends using PODYSPEC for climate change contexts: "For applications in a climate change context, the PODYSPEC method is recommended as key factors such as phenology and soil moisture are not included in the parameterization of PODYIAM."

> **Text Box 9: Applications for vegetation-type flux models and critical levels, POD$_Y$IAM**
>
> These flux models have simpler form than POD$_Y$SPEC and have been developed specifically for use in large-scale integrated assessment modelling, including for scenario analysis and optimisation runs. Separate parametrisations are provided for Mediterranean and non-Mediterranean areas for application in risk assessments for crops, forest trees and (semi-)natural vegetation.
>
> The flux-effect relationships can be used for:
>
> - **Crops:** potential maximum yield loss calculation and indicative economic losses in worst case scenario;
> - **Forest trees and (semi-)natural vegetation:** indicative of the potential maximum risk for estimating environmental cost, but not economic losses.
>
> The critical levels can be used for calculating critical levels exceedances, both amount and area. For applications in a climate change context, the POD$_Y$SPEC method is recommended as key factors such as phenology and soil moisture are not included in the parameterisation of POD$_Y$IAM.

The PODYIAM model is indeed a simplification of more detailed flux models (such as those based on PODYSPEC), suitable only when the data required for PODYSPEC application is unavailable. Additionally, PODYIAM does not specify any particular plant species, unlike our study, that is specifically designed to target wheat.

Finally, we would like to highlight that PODYSPEC for wheat and PODYIAM for crops have precisely the same parameterization (see the table reported below, LRTAP Convention, 2017),

with the sole exception that PODYIAM does not account for soil moisture or phenology—factors that are indeed essential in the context of climate change.

| POD3IAM definition | POD6SPEC definition |
|---|---|

**POD3IAM definition**

*Table III.15: Parameterisation of the DO$_3$SE model for POD$_Y$IAM calculat... forests and (semi-) natural vegetation. Separate parameterisations...*

| Parameter | Units | Crop parameterisation POD$_3$IAM | | Fore... |
|---|---|---|---|---|
| Biogeographic region | | Atlantic, Boreal, Continental, Steppic, Pannonian | Mediterranean | At... Cont... |
| Based on species | | Wheat | Wheat | Beech... |
| $g_{max}$ | mmol O$_3$ m$^{-2}$ PLA s$^{-1}$ | 500 | 430 | |
| $f_{min}$ | fraction | 0.01 | 0.01 | |
| light_a | - | 0.0105 | 0.0105 | |
| $T_{min}$ | °C | 12 | 13 | |
| $T_{opt}$ | °C | 26 | 28 | |
| $T_{max}$ | °C | 40 | 39 | |
| VPD$_{max}$ | kPa | 1.2 | 3.2 | |
| VPD$_{min}$ | kPa | 3.2 | 4.6 | |
| ΣVPD$_{crit}$ | kPa | 8 | 8 | |
| PAW$_t$ | % | $f_{SW} = 1$ | $f_{SW} = 1$ | |
| SWC$_{max}$ | % volume | - | - | |

**POD6SPEC definition**

| Parameter | Units | Crop species parameterisa... | | |
|---|---|---|---|---|
| Region (may also be applicable in these regions) | | Atlantic, Boreal, Continental (Pannonian, Steppic) | Mediterranean | Mediterranean |
| Species — Common name | | (Bread) Wheat | (Bread) Wheat | (Durum) Wheat |
| Species — Latin name | | *Triticum aestivum* | *Triticum aestivum* | *Triticum durum* |
| $g_{max}$ | mmol O$_3$ m$^{-2}$ PLA s$^{-1}$ | 500 | 430 | 410 |
| $f_{min}$ | fraction | 0.01 | 0.01 | 0.01 |
| light_a | - | 0.0105 | 0.0105 | 0.0105 |
| $T_{min}$ | °C | 12 | 12 | 11 |
| $T_{opt}$ | °C | 26 | 28 | 28 |
| $T_{max}$ | °C | 40 | 39 | 45 |
| VPD$_{max}$ | kPa | 1.2 | 3.2 | 3.1 |
| VPD$_{min}$ | kPa | 3.2 | 4.6 | 4.9 |
| ΣVPD$_{crit}$ | kPa | 8 | 16 | 16 |
| PAW$_t$[i] | % | 50 | - | - |
| SWC$_{max}$[i] | % volume | - | 18.6 | 18.0 |

**4. For these reasons the global scale POD assessments of Mills et al. (2018b,c) made use of POD3IAM metric. And although neither of the Mills papers was able to evaluate even this POD3 metric globally, they did show that the EMEP chemical transport model that was used was able to satisfactorily reproduce some basic statistics, namely mean of daily maximum ozone and M7, at sites from around the globe (Mills et al., 2018b, SI), and that model had been extensively tested against field data relevant to ozone deposition and fluxes.**

Mills et al. (2018a, b) assessed metrics relevant to TOAR-I. In the current TOAR-II framework, the Ozone Deposition Focus working group focuses on metrics such as PODY, as will be seen in other papers that will be submitted to the TOAR-II special issue.

Regarding the reviewer's claim that the models were extensively tested against field data relevant to ozone deposition and fluxes, this deserves further discussion. As we understand it, the EMEP model reproduces ozone fluxes by coupling with the DO3SE model (Simpson et al., 2012). If, as the reviewer suggests, this model reliably reproduces ozone deposition and fluxes, our deposition model—which is based on the DO3SE—should also be capable of doing so, and this was demonstrated even at fine temporal resolution as reported in the previously cited comparison exercise described in the supplementary materials.

To our knowledge, no published studies directly compare the ozone fluxes or POD metrics predicted by the DO3SE model against direct measurements of ozone fluxes in crop fields. If the reviewer is aware of such studies, we welcome references.

For instance, Mills et al. (2018b) compared satellite-estimates of evapotranspiration (i.e. latent heat flux, LE) in the US with the modelled POD3IAM, under the assumption that both POD3IAM and evapotranspiration are driven by stomatal conductance. However, their approach presents clear limitations: (i) there was no discussion on the temporal resolution of the satellite measurements; (ii) assuming 10% wheat cover in a grid cell does not ensure representativeness of the grid cell's ET for wheat's water exchange; (iii) using three-year averages excessively smooths and ease the comparison; (iv) the comparison was indirect, making claims of validation difficult. Therefore, we decided to directly validate our model against flux measurements obtained over a wheat field (the aim of this manuscript) with the eddy covariance technique and show that our model satisfactorily represents LE fluxes on a half-hourly basis. Please refer to the supplementary materials for detailed analyses and results.

**5. The usual problems of accurately modeling O3 and its metrics are exacerbated when climate models are used. In this case the meteorology is not constrained by reanalysis, and hence diverges more from the real-world than usually seen in current day chemical transport models. So, how well can your models predict O3, M7, and AOT40 for example (ie the metrics which can be derived from global observations), and indeed the hourly frequency distribution of O3?**

The reviewer suggests that meteorology might be distorted in climate models due to the lack of reanalysis constraints, potentially affecting the prediction of photochemical ozone production. While this could be theoretically valid, CMIP climate models undergo extensive testing and continuous refinement to address meteorological distortions, both in terms of biases and in terms of trends. At each stage of CMIP model development, evaluations of key meteorological variables (including biases, trends, and seasonal variability) is routinely conducted. These assessments are consistently reported in the literature associated with CMIP model documentation, to which we refer the reviewer (e.g., Dunne et al., 2020; Horowitz et al., 2020; Sellar et al., 2019). Furthermore, it is important to mention that in order to make future projections of ozone damage, it is not possible to use reanalysis products, but climate models are the only possibility, even with the obvious uncertainties attached.

In the supplementary material attached to this reply we present an evaluation of the $O_3$ bias only for the daylight hours and over the accumulation period for POD (please refer to it), and show that the O3 concentration bias is small on average, and that, even for the regions where the bias is larger (such as East Asia), the effect on the $POD_6$ is relatively small.

**6. A related issue is also that this paper seems to use quite short slices of meteorology. The base simulation is for 15 years (2000-2014), and the climate runs seem to be for 10 years (though I am a little confused by the 10 year slices given on L134 and the 15-year slice mentioned on L116). With short time-slices there is an increased risk that changes**

seen are due to random variations rather than to a true climate signal. Even with 20 year time-slices Langner et al. (2012) showed that the changes seen in summertime ozone were not significant at the 95% level over large parts of Europe.

The 10-year time slices used for projecting variable fields in 2050 and 2100 are an established approach in climate change studies, especially for air quality studies, as supported by several papers (e.g., Griffiths et al., 2020; Ronan et al., 2020; Sellar et al., 2019; Turnock et al., 2020). Furthermore, a 10-years slice seems adequate to capture the rate of change of precursor emissions. A 30-years average, while being a conventional averaging period for climate, will not be appropriate to reflect the rapid changes in ozone precursor emissions. In any case, the interpretation of the regional POD changes, the shown spatial and temporal trends for $POD_6$ over the century (Figure 4), and the statistical significance of the results (Table 4), indicate that a signal exists and that the shown trends reflect real tendencies rather than random fluctuations.

In any case, following the comments in the review of Owen Cooper, we added specific p-values and confidence intervals in the manuscript (section 3.3) where appropriate, in order to point out uncertain results. Furthermore, the supplementary materials will contain, in their final form, a table with p-values and 95% confidence intervals corresponding to table 4 in the manuscript, and a map of the p-value associated with the ANOVA (i.e. Figure 8).

**7. In the manuscript here, there is no discussion of these key issues. Instead we are referred to Turnock et al. (2020) for information about model skill, but that paper states that "CMIP6 models consistently overestimate observed surface O3 concentrations across most regions and in most seasons by up to 16 ppb, with a large diversity in simulated values over Northern Hemisphere continental regions".**

The "up to 16 ppb" value the reviewer mentions indicates the maximum mean bias, and thus does not apply universally across all stations. While CMIP6 models are found to generally overestimate $O_3$ (Turnock et al., 2020), this overestimation is neither uniform nor pervasive across all models, and moreover does not account for the deposition processes that we consider in our study (please see the new supplementary materials attached below). As mentioned above, an $O_3$ bias evaluation tailored to our case-study is provided now in the supplementary materials, where $O_3$ concentrations are found to be modelled adequately for the purposes of our study.

**8. Given such issues with surface O3 and the modeling in general, I have no reason to believe that the POD6 values have satisfactory values, or that trends in this metric are any more reliable.**

In conclusion, we emphasize that the limitations and uncertainties of our study were adequately reported and quantified in the manuscript, and that the evaluation requested by the reviewer does not affect the validity of our study nor its conclusions. Following the reviewer's request, we performed an additional evaluation of the $O_3$ concentrations simulated by the CMIP6 models, which represents an extension of was what performed in Turnock et al. (2020), and we also tested the performances of our deposition model in reproducing the half-hourly latent heat flux.

Regarding the choice of the $POD_6$ metric, we believe that the $POD_6$ metric is the most appropriate for climatic studies like ours, as supported by both the mapping manual (referenced by the reviewer, though) and the cited relevant literature.

In any case, we want to highlight that combining different approaches (i.e., exposure-based approaches in most earlier studies and dose-based approach in this study) in assessing $O_3$ damage on vegetation may better constrain the uncertainties in the modeled results, as has been demonstrated in ensemble modeling air-quality and climate studies in literature. Therefore, since existing dose-based studies of $O_3$ damage on vegetation are very rare, our study is an important contribution to the current knowledge, despite known model uncertainties mentioned above.

**Other comments**

**p2, L48. Strange not to mention the Mills et al. papers here, or that of Van Dingenen et al., 2009; these papers both offer both global-scale assessments which involved a lot of work (including comparison with observations) and are widely cited.**

Thanks for the suggestion. We read the papers and the citations were added.

**p4, Table 1. This table should also include the thickness of the lowest model, as this is the important for deriving crop-height O3 concentrations.**

Yes. The height of the lowest model level (cell centre) was added in the table

**p5. on L116, we read that the baseline is calculated for 15 years, over 2000-2014. How many years are used for the 2100 simulations?**

The POD6 was calculated every year from 2015 to 2100. Each individual year was compared to the baseline average. To avoid possible confusion, the beginning of that sentence has been rephrased as: "Yearly $O_3$ risk for wheat cultivation from 2015 to 2100 is quantified with respect to a POD baseline value,…"

**p5, L139. What does "Contextually" mean here? It sounds odd.**

Thanks. "Contextually" was substituted with "moreover".

**P5. Sect.2.2 The text suggests that wilting point and field capacity are needed, and derived as volumetric soil moisture (VSM) values. One issue is that the ESMs will have their own systems for dealing with soil water, and their calculations of near-surface ozone and resistances in general will presumably reflect their interpretation of soil-water effects. Possibly more serious is the use of volumetric soil water (VSM). The same VSM can represent very wet conditions in some soils, but very dry in others, but as far as I can tell the methods don't distinguish between different soils at all.**

Indeed, the ESMs do represent soil water and offer the corresponding output. However, the soil water values from the ESMs at a given node are influenced by various land covers and land uses within both the grid node itself and its surrounding areas, and for this reason they are not intended to be referred to a single wheat field, but rather to the average soil water content of a large area. On the contrary, we wanted to simulate the soil water content only in a wheat field, without taking into account other land covers. Therefore our deposition model uses an online water soil module, following the approach of Mintz and Walker (1993). This is described in detail in our previous paper (Guaita et al., 2023) that we refer to.

Further, we note that different soil types have different associated wilting point and field capacity values. Therefore, these parameters effectively distinguish between soil types (please, see the dataset referenced in the manuscript, Zhang et al., 2018).

**p6. The text here omits any mention of the difference between leaf and canopy scale resistances, but this is a key part of the DO3SE methodology (e.g. Tuovinen et al., 2009)**

Our model uses the same approach as the DO3SE methodology. This was thoroughly discussed in the paper Guaita et al., 2023, which is frequently cited across the manuscript (Please see eq. 46-52 in Guaita et al., 2023).

According to the reviewer's remark, in the revised paper we clarify the difference between the bulk resistances (upcase R) used to scale the ozone concentration from the lowest model level to the canopy height, and the leaf-level resistances (lowcase r), which are used to calculate the ozone stomatal flux for an upper canopy leaf as prescribed by the DO3SE methodology.

**p7, L173. Again the word contextually is used. It fits better here than in the above example, but I think it is better to say "In the context of…".**

Thanks, the sentence was modified as suggested.

**p7, L174. The word parameterizations is a bit vague, and readers cannot be expected to know what this means. Please make a table with the parameter values.**

Ok, we have now included a table in the supplementary materials which we refer to in the main manuscript.

**p7, L184. In what way are the Feng et al. (2012) parameterizations incomplete? I would have thought that methods developed from China were more appropriate for global approaches than those from Spain.**

The parameterization proposed by Feng et al. (2012) did not consider limitations to stomatal conductance from temperature and soil water content. Since these variables are important in a climate change context, we preferred to exclude this parameterization in our study. In fact, Feng et al. (2012) adopted the following formulation for the Jarvis model:

**2.3. The multiplicative stomatal conductance model**

The present study used a multiplicative stomatal conductance algorithm adapted from LRTAP (2010):

$$g_{sto} = g_{max} \times \min\left(f_{phen}, f_{O_3}\right) \times f_{light} \times \max(f_{min}, f_{VPD}), \qquad (1)$$

Since temperature and soil moisture are important in a climate change context, in our study we preferred to exclude the parameterization of Feng et al. (2012). On the contrary, the parameterization from González-Fernández et al. (2013) includes the effect of soil water and temperature as well, and was based on field measurements datasets.

**p7, L195 "A well-established dose-response relationship...". Is this so well established? The mapping manual states "the percentage effect due to O3 impact on crop yield estimated in large-scale modeling should be calculated as follows:**

 **(PODYIAM – Ref10 PODYIAM) * (% reduction per mmol/m2 PODYIAM.POD3)**

**And indeed, Mills et al (2018c) used:**

  **RYL = (POD3IAM-0.1)*0.64**

**but this manuscript uses POD6 rather than the recommended POD3IAM for unexplained reasons, and makes no mention of the "Ref10" correction.**

We removed the adjective "well-established". However, as discussed in the major comments above, POD6 is indeed the right approach in our case (LRTAP Convention, 2017, pag. 45). Following the mapping manual, the relative yield loss in this case is given by:

RYL = (PODYSPEC – Ref10 PODYSPEC) * % reduction per mmol m-2 PODYSPEC

However, the Ref10 PODYSPEC is null in the case of wheat (see the Table III.10 of the Mapping Manual, LRTAP Convention, 2017, reported below), and therefore it was omitted.

*Table III.10*: $POD_6SPEC$ critical levels (CL) for crops.

| Species | Effect parameter | Biogeo-graphical region* | Potential effect at CL (% reduction) | Critical level (mmol $m^{-2}$ PLA)** | Ref10 $POD_6$ (mmol $m^{-2}$ PLA) | Potential maximum rate of reduction (%) per mmol $m^{-2}$ PLA of $POD_6SPEC$*** |
|---|---|---|---|---|---|---|
| Wheat | Grain yield | A,B,C,M (S,P)**** | 5% | 1.3 | 0.0 | 3.85 |
| Wheat | 1000-grain weight | A,B,C,M (S,P)**** | 5% | 1.5 | 0.0 | 3.35 |
| Wheat | Protein yield | A,B,C,M (S,P)**** | 5% | 2.0 | 0.0 | 2.54 |
| Potato | Tuber yield | A,B,C (M,S,P) | 5% | 3.8 | 0.0 | 1.34 |
| Tomato | Fruit yield | M (A,B, C,S,P) | 5% | 2.0 | 0.0 | 2.53 |
| Tomato | Fruit quality | M (A,B, C,S,P) | 5% | 3.8 | 0.0 | 1.30 |

**P7, L202—204. This sentence seems out of place compared to the preceding text.**

Yes. The sentence was moved to the Appendix B, and it was referenced within the section 2.3.

**p24, L456. Given all the uncertainties I mentioned in the major comments section, I wonder what the phrase "Our results may be associated with different degrees of confidence depending on the agreement between the two available CMIP6 models," means? The paper has barely mentioned the main sources of uncertainty I think.**

Different physics-based models displaying similar features suggest that their results are associated with a greater degree of confidence, as opposed to the two models disagreeing. Of course, considering more models would improve the study, but the UKESM1-0-LL and GFDL-ESM4 remain to date the only models within CMIP6 that are suitable for our study. This is now clarified in the manuscript.

As shown in the newly written supplementary materials, the effect of the O3 bias on POD6 is found to be negligible for average climatic conditions in two of the main areas impacted by O3 risk at the present time (North America and Europe). As for the other region (East Asia), the $O_3$ bias could lead to an 0.28-0.33 mmol $m^{-2}$ overestimation of $POD_6$ on average, which however would not be enough to classify this region as not at risk, and therefore such possible overestimation does not affect our main conclusions. A discussion on this has been added to the discussion section.

**p42, L890 UKESM  - 20m. Is that cell depth, or cell-center?**

It's the cell-center. Thanks. This has been added to the Appendix.

**References from authors**

Dunne, J. P., Horowitz, L. W., Adcroft, A. J., Ginoux, P., Held, I. M., John, J. G., Krasting, J. P., Malyshev, S., Naik, V., Paulot, F., Shevliakova, E., Stock, C. A., Zadeh, N., Balaji, V., Blanton, C., Dunne, K. A., Dupuis, C., Durachta, J., Dussin, R., Gauthier, P. P. G., Griffies, S. M., Guo, H., Hallberg, R. W., Harrison, M., He, J., Hurlin, W., McHugh, C., Menzel, R., Milly, P. C. D., Nikonov, S., Paynter, D. J., Ploshay, J., Radhakrishnan, A., Rand, K., Reichl, B. G., Robinson, T., Schwarzkopf, D. M., Sentman, L. T., Underwood, S., Vahlenkamp, H., Winton, M., Wittenberg, A. T., Wyman, B., Zeng, Y., and Zhao, M.: The GFDL Earth System Model Version 4.1 (GFDL-ESM 4.1): Overall Coupled Model Description and Simulation Characteristics, J Adv Model Earth Syst, 12, e2019MS002015, https://doi.org/10.1029/2019MS002015, 2020.

Emberson, L.: Effects of ozone on agriculture, forests and grasslands, Phil. Trans. R. Soc. A., 378, 20190327, https://doi.org/10.1098/rsta.2019.0327, 2020.

Feng, Z., Tang, H., Uddling, J., Pleijel, H., Kobayashi, K., Zhu, J., Oue, H., and Guo, W.: A stomatal ozone flux–response relationship to assess ozone-induced yield loss of winter wheat in subtropical China, Environmental Pollution, 164, 16–23, https://doi.org/10.1016/j.envpol.2012.01.014, 2012.

Gerosa, G., Cieslik, S., and Ballarin-Denti, A.: Micrometeorological determination of time-integrated stomatal ozone fluxes over wheat: a case study in Northern Italy, Atmospheric Environment, 37, 777–788, https://doi.org/10.1016/S1352-2310(02)00927-5, 2003.

González-Fernández, I., Bermejo, V., Elvira, S., De La Torre, D., González, A., Navarrete, L., Sanz, J., Calvete, H., García-Gómez, H., López, A., Serra, J., Lafarga, A., Armesto, A. P., Calvo, A., and Alonso, R.: Modelling ozone stomatal flux of wheat under mediterranean conditions, Atmospheric Environment, 67, 149–160, https://doi.org/10.1016/j.atmosenv.2012.10.043, 2013.

Griffiths, P. T., Murray, L. T., Zeng, G., Archibald, A. T., Emmons, L. K., Galbally, I., Hassler, B., Horowitz, L. W., Keeble, J., Liu, J., Moeini, O., Naik, V., O'Connor, F. M., Shin, Y. M., Tarasick, D., Tilmes, S., Turnock, S. T., Wild, O., Young, P. J., and Zanis, P.: Tropospheric ozone in CMIP6 Simulations, https://doi.org/10.5194/acp-2019-1216, 10 February 2020.

Guaita, P. R., Marzuoli, R., and Gerosa, G. A.: A regional scale flux-based O3 risk assessment for winter wheat in northern Italy, and effects of different spatio-temporal resolutions, Environmental Pollution, 333, 121860, https://doi.org/10.1016/j.envpol.2023.121860, 2023.

Horowitz, L. W., Naik, V., Paulot, F., Ginoux, P. A., Dunne, J. P., Mao, J., Schnell, J., Chen, X., He, J., John, J. G., Lin, M., Lin, P., Malyshev, S., Paynter, D., Shevliakova, E., and Zhao, M.: The GFDL Global Atmospheric Chemistry-Climate Model AM4.1: Model Description and Simulation Characteristics, J Adv Model Earth Syst, 12, e2019MS002032, https://doi.org/10.1029/2019MS002032, 2020.

LRTAP Convention: Chapter III: mapping critical level for vegetation, in: Modelling and Mapping Manual, 2017.

Mills, G., Hayes, F., Simpson, D., Emberson, L., Norris, D., Harmens, H., and Büker, P.: Evidence of widespread effects of ozone on crops and (semi-)natural vegetation in Europe (1990-2006) in relation to AOT40- and flux-based risk maps: OZONE EFFECTS ON VEGETATION IN EUROPE, Global Change Biology, 17, 592–613, https://doi.org/10.1111/j.1365-2486.2010.02217.x, 2011.

Mills, G., Sharps, K., Simpson, D., Pleijel, H., Frei, M., Burkey, K., Emberson, L., Uddling, J., Broberg, M., Feng, Z., Kobayashi, K., and Agrawal, M.: Closing the global ozone yield gap: Quantification and cobenefits for multistress tolerance, Global Change Biology, 24, 4869–4893, https://doi.org/10.1111/gcb.14381, 2018a.

Mills, G., Sharps, K., Simpson, D., Pleijel, H., Broberg, M., Uddling, J., Jaramillo, F., Davies, W. J., Dentener, F., Van Den Berg, M., Agrawal, M., Agrawal, S. B., Ainsworth, E. A., Büker, P., Emberson, L., Feng, Z., Harmens, H., Hayes, F., Kobayashi, K., Paoletti, E., and Van Dingenen, R.: Ozone pollution will compromise efforts to increase global wheat production, Global Change Biology, 24, 3560–3574, https://doi.org/10.1111/gcb.14157, 2018b.

Mintz, Y. and Walker, G. K.: Global Fields of Soil Moisture and Land Surface Evapotranspiration Derived from Observed Precipitation and Surface Air Temperature, J. Appl. Meteor., 32, 1305–1334, https://doi.org/10.1175/1520-0450(1993)032<1305:GFOSMA>2.0.CO;2, 1993.

Ronan, A. C., Ducker, J. A., Schnell, J. L., and Holmes, C. D.: Have improvements in ozone air quality reduced ozone uptake into plants?, Elem Sci Anth, 8, 2, https://doi.org/10.1525/elementa.399, 2020.

Schröder, Sabine; Schultz, Martin G.; Selke, Niklas; Sun, Jianing; Ahring, Jessica; Mozaffari, Amirpasha; Romberg, Mathilde; Epp, Eleonora; Lensing, Max; Apweiler, Sander; Leufen, Lukas H.; Betancourt, Clara; Hagemeier, Björn; Rajveer, Saini: TOAR Data Infrastructure, https://doi.org/10.34730/4D9A287DEC0B42F1AA6D244DE8F19EB3, 2021.

Sellar, A. A., Jones, C. G., Mulcahy, J. P., Tang, Y., Yool, A., Wiltshire, A., O'Connor, F. M., Stringer, M., Hill, R., Palmieri, J., Woodward, S., De Mora, L., Kuhlbrodt, T., Rumbold, S. T., Kelley, D. I., Ellis, R., Johnson, C. E., Walton, J., Abraham, N. L., Andrews, M. B., Andrews, T., Archibald, A. T., Berthou, S., Burke, E., Blockley, E., Carslaw, K., Dalvi, M., Edwards, J., Folberth, G. A., Gedney, N., Griffiths, P. T., Harper, A. B., Hendry, M. A., Hewitt, A. J., Johnson, B., Jones, A., Jones, C. D., Keeble, J., Liddicoat, S., Morgenstern, O., Parker, R. J., Predoi, V., Robertson, E., Siahaan, A., Smith, R. S., Swaminathan, R., Woodhouse, M. T., Zeng, G., and Zerroukat, M.: UKESM1: Description and Evaluation of the U.K. Earth System Model, J Adv Model Earth Syst, 11, 4513–4558, https://doi.org/10.1029/2019MS001739, 2019.

Simpson, D., Benedictow, A., Berge, H., Bergström, R., Emberson, L. D., Fagerli, H., Flechard, C. R., Hayman, G. D., Gauss, M., Jonson, J. E., Jenkin, M. E., Nyíri, A., Richter, C., Semeena, V. S., Tsyro, S., Tuovinen, J.-P., Valdebenito, Á., and Wind, P.: The EMEP MSC-W chemical transport model – technical description, Atmos. Chem. Phys., 12, 7825–7865, https://doi.org/10.5194/acp-12-7825-2012, 2012.

Turnock, S. T., Allen, R. J., Andrews, M., Bauer, S. E., Deushi, M., Emmons, L., Good, P., Horowitz, L., John, J. G., Michou, M., Nabat, P., Naik, V., Neubauer, D., O'Connor, F. M., Olivié, D., Oshima, N., Schulz, M., Sellar, A., Shim, S., Takemura, T., Tilmes, S., Tsigaridis, K., Wu, T.,

and Zhang, J.: Historical and future changes in air pollutants from CMIP6 models, Atmos. Chem. Phys., 20, 14547–14579, https://doi.org/10.5194/acp-20-14547-2020, 2020.

Zhang, Y., Schaap, M. G., and Zha, Y.: A High-Resolution Global Map of Soil Hydraulic Properties Produced by a Hierarchical Parameterization of a Physically Based Water Retention Model, Water Resources Research, 54, 9774–9790, https://doi.org/10.1029/2018WR023539, 2018.

**References from reviewer**

Langner, J., et al., European summer surface ozone 1990–2100, Atmos. Chem. Physics, 12, 10097–10105, https://doi.org/10.5194/acp-12-10097-2012, 2012.

Lefohn, A. S., et al., Tropospheric ozone assessment report: Global ozone metrics for climate change, human health, and crop/ecosystem research., Elem Sci Anth., 6, 28, https://doi.org/http://doi.org/10.1525/elementa.279, 2018.

Mills, G., et al., Tropospheric Ozone Assessment Report: Present-day tropospheric ozone distribution and trends relevant to vegetation., Elem. Sci. Anth., 6, https://doi.org/10.1525/elementa.302, 2018a.

Mills, G., et al., Ozone pollution will compromise efforts to increase global wheat production, Global Change Biol., 24, 3560–3574, https://doi.org/10.1111/gcb.14157, 2018b.

Mills, G., et al., Closing the global ozone yield gap: Quantification and cobenefits for multistress tolerance, Global Change Biol., https://doi.org/10.1111/gcb.14381, 2018c.

Sofiev, M. and Tuovinen, J.-P.: Factors determining the robustness of AOT40 and other ozone exposure indices, Atmos. Environ., 35, 3521–3528, 2001.

Tuovinen, J.-P.: Assessing vegetation exposure to ozone: properties of the AOT40 index and modifications by deposition modelling, Environ. Poll., 109, 361–372, 2000.

Tuovinen, J.-P., et al., Robustness of modelled ozone exposures and doses, Environ. Poll., 146, 578–586, 2007.

Tuovinen, J.-P., et al., Modelling ozone fluxes to forests for risk assessment: status and prospects, Annals of Forest Science, 66, 401, 2009.

Turnock, S. T., et al., Historical and future changes in air pollutants from CMIP6 models, Atmos. Chem. Physics, 20, 14547–14579, https://doi.org/10.5194/acp-20-14547-2020, 2020.

Van Dingenen, R., et al, The global impact of ozone on agricultural crop yields under current and future air quality legislation, Atmos. Environ., 43, 604–618, https://doi.org/10.1016/j.atmosenv.2008.10.033, 2009.

Supplement of

**Global flux-based ozone risk assessment for wheat up to 2100 under different climate scenarios**

Pierluigi R. Guaita[1,8], Riccardo Marzuoli[1], Leiming Zhang[2], Steven Turnock[3,4], Gerbrand Koren[5], Oliver Wild[6], Paola Crippa[7], Giacomo Gerosa[1]

[1]Dep. Mathematics and Physics, Catholic University of the Sacred Heart, Brescia, Italy
[2]Air Quality Research Division, Science and Technology Branch, Environment and Climate Change Canada, Toronto, Canada
[3]Met Office Hadley Center, Exeter, UK
[4]University of Leeds Met Office Strategic (LUMOS) Research Group, University of Leeds, UK
[5]Copernicus Institute of Sustainable Development, Utrecht University, Utrecht, The Netherlands
[6]Lancaster Environment Centre, Lancaster University, Lancaster, UK
[7]Department of Civil and Environmental Engineering and Earth Sciences, University of Notre Dame, Notre Dame, IN, USA
[8]Department of Applied Computational Mathematics and Statistics, University of Notre Dame, Notre Dame, IN, USA

*Correspondence to*: Pierluigi R. Guaita (pierluigirenan.guaita@unicatt.it), Giacomo A. Gerosa (giacomo.gerosa@unicatt.it)

**1. O₃ mean bias and its effect on POD₆**

The bias of the $O_3$ concentrations (Table S1, Figure S1) taken as input data by our dry deposition model are evaluated against the surface $O_3$ observations taken from the TOAR-II dataset (Schröder et al., 2021) for the baseline years (2000-2014). For each measurement station, the mean bias (MB) is defined as:

$$MB(x) = \frac{\sum_{t=1}^{n} M(x,t) - O(x,t)}{n}$$

(S1)

where $x$ indicates the location, and $t = 1, \dots, n$ are the considered timesteps (daylight hours, over the accumulation period for that location). $M(x,t)$ is the $O_3$ output from the considered CMIP6 models (sfo3 in CMIP6 notation, indicating the $O_3$ at the lowest model level), scaled at the canopy height by means of the dry deposition scheme used in this study (Guaita et al., 2023), and $O(x,t)$ is the observed surface $O_3$ concentration (also scaled to canopy height). The ground stations are selected to be representative of the agricultural context, and therefore only rural, background or crop locations are included. Since the dry deposition model of this study scales $O_3$ from the height of the lowest model level of the CMIP6 models to the crop canopy height, only ground stations with measurements height below 3 meters are considered. Furthermore, only stations with more than 10 years long timeseries are included. In this regard, an exception is made for China, as the stations that could represent wheat fields have timeseries in that region are shorter than 10 years.

To estimate the effect of the $O_3$ bias on the $O_3$ risk, the yearly POD₆ was regressed (with interaction) against $O_3$ concentration and $f_{clim}$ averaged over the accumulation period:

$$POD_6 = \beta_0 + \beta_1[O_3] + \beta_2 f_{clim} + \beta_3[O_3]f_{clim} + \epsilon$$

(S2)

where $\beta_i$ ($i = 0, \dots, 3$) are the regression coefficients, and $\epsilon \in N(0, \sigma^2)$ is the error. The regression is calibrated over the whole globe, using both the baseline and the three scenarios considered in this study (SSP1-2.6, SSP3-7.0, SSP5-8.5). In the eq. (S2), $\beta_1 + \beta_3 \cdot f_{clim}$ is the mean POD₆ increment per ppb given a certain value of $f_{clim}$. Therefore, the expression $(\beta_1 + \beta_3 \cdot f_{clim}) \cdot MB$ represents the effect of the $O_3$ MB on the POD₆. In other words, when $f_{clim}$ is equal to the regional (or global) average ($\mu(f_{clim})$), the resulting value is the average effect of the $O_3$ MB on the POD₆ over the specified region under mean $f_{clim}$ conditions, while, when $f_{clim} = 1$, the resulting value is the average MB effect on POD₆ under climatic conditions optimal to the stomatal conductance (Table S1).

**Table S1: MB of O₃ at canopy height, POD₆ increments per ppb of O₃ concentration conditional to $f_{clim}$ equal to its regional mean ($\beta_1 + \beta_3 \cdot \mu(f_{clim})$), and to $f_{clim} = 1$ ($\beta_1 + \beta_3 \cdot 1$), and the corresponding projected effect on POD₆ for average climatic conditions and optimal conditions to the stomatal conductance.**

| Region | # station | MB ± SD [ppb] | $\beta_1 + \beta_3 \cdot \mu(f_{clim})$ (95%CI) [mmol m$^{-2}$/ppb] | $\beta_1 + \beta_3 \cdot 1$ (95%CI) [mmol m$^{-2}$/ppb] | $(\beta_1 + \beta_3 \cdot \mu(f_{clim})) \cdot$ MB ± SD [mmol m$^{-2}$] | $(\beta_1 + \beta_3 \cdot 1) \cdot$ MB ± SD [mmol m$^{-2}$] |
|---|---|---|---|---|---|---|
| **GFDL-ESM4** | | | | | | |
| Global | 1330 | 1.31 ± 7.36 | 0.0154 (0.0153,0.0155) | 0.0582 (0.0581,0.0584) | 0.02±0.03 | 0.08±0.37 |
| N. America | 579 | -2.8 ± 4.96 | 0.0188 (0.0186,0.0189) | 0.0493 (0.049,0.0496) | -0.05±0.02 | -0.14±0.14 |
| Europe | 480 | 2.09 ± 5.59 | 0.0152 (0.015,0.0154) | 0.0573 (0.0567,0.0579) | 0.03±0.02 | 0.12±0.22 |
| East Asia | 283 | 8.02 ± 8.53 | 0.0351 (0.0348,0.0354) | 0.0913 (0.0907,0.0918) | 0.28±0.26 | 0.73±1.76 |
| **UKESM1-0-LL** | | | | | | |
| Global | 1350 | 0.88 ± 7.61 | 0.0316 (0.0313,0.0318) | 0.1192 (0.1186,0.1198) | 0.03±0.12 | 0.1±1.67 |
| N. America | 579 | -3.27 ± 5.91 | 0.0441 (0.0436,0.0446) | 0.1289 (0.1276,0.1302) | -0.14±0.16 | -0.42±1.35 |
| Europe | 486 | 3.13 ± 5.58 | 0.0401 (0.0393,0.0409) | 0.1614 (0.1579,0.165) | 0.13±0.12 | 0.51±1.93 |
| East Asia | 276 | 5.59 ± 9.34 | 0.0588 (0.058,0.0597) | 0.164 (0.162,0.1659) | 0.33±0.72 | 0.92±5.61 |

[Figure]

[Figure]

**Figure S1: MB of O₃ at canopy height for GFDL-ESM4 (a), and UKESM1-0-LL (b), obtained by scaling O₃ in output from the CMIP6 models (at the lowest model level height), to the canopy height, by means of the resistive network of the dry deposition model. O₃ is compared with ground measurements from the TOAR-II database.**

**2. Evaluation of the performance of the dry deposition model used for POD₆ calculations**

The dry deposition model used in this study (Guaita et al., 2023) is compared to the ozone flux measurements made on a wheat field wheat in Comun Nuovo (Italy) (Gerosa et al., 2003).

The model is tested for its capability to reproduce the total O₃ flux ($F_{O3}$) over the wheat field which corresponds to testing the resistance network altogether, and the latent heat flux (LE, W/m$^2$) which is a proxy for stomatal conductance (and therefore a proxy for stomatal resistance). Namely, the total stomatal flux is calculated as follows:

$$F_{O3} = \frac{O_3(z_{mO3})}{R_{aH}(d+z_{0m})+R_{bO3}+R_{surf,O3}} \tag{S3}$$

where $z_{mO3}$ is the measurement height, $d$ is the displacement height, $z_{0m}$ is the roughness length for momentum, $O_3(z_{mO3})$ is the ozone concentration at measured height, $R_{aH}(d + z_{0m}, z_{mO3})$ is the aerodynamic resistance between $d + z_{0m}$ and $z_{mO3}$, $R_{bO3}$ is the quasi-laminar resistance, and $R_{surf,O3}$ is the bulk overall surface resistance to deposition. For details on the calculation for each of these variables, see the Appendix A in Guaita et al. (2023).

Figure S2 shows the timeseries of $F_{O3}$ and LE from the beginning of flowering to the harvest. Table S2 shows statistics for the model performance over the daylight hours (6 am-6 pm). The reported values are obtained by simply regressing the modelled values against the observed ones.

[Figure]

**Figure S2: Timeseries for modelled and observed $F_{O3}$ (a), and LE (b)**

**Table S2: Statistics for the comparison between observed and measured $F_{O3}$, and LE. The regression uses the modelled data as predictand and the observations as predictor.**

|  | $F_{O3}$ (nmol m$^{-2}$ s$^{-1}$) | LE (W m$^{-2}$) |
|---|---|---|
| Regression intercept (p-value) | 4.3377 (1.8448e-24) | -0.0209 (3.5875e-14) |
| Regression slope (p-value) | 1.0481 (1.3058e-101) | 1.1438 (1.0069e-197) |
| Mean Bias | 3.3298 | -0.0089 |
| Mean Absolute Error | 4.0317 | 0.0200 |
| Root Mean Square Error | 5.8807 | 0.0308 |
| R-squared | 0.4965 | 0.7403 |

---

## Author Comment (AC4)

**Table S4: p-values and 95% confidence intervals (corrected with Bonferroni) corresponding to the differences indicated in Table 4. Boundaries for p-values are defined according to the TOAR-II Recommendations for Statistical Analysis (Chang et al. 2023). $p<0.01$ indicates very high certainty and $p>0.33$ indicates very low certainty or no evidence.**

| Region[b] | 2050 $\Delta POD_6$ [mmol m$^{-2}$] | | | 2100 $\Delta POD_6$ [mmol m$^{-2}$] | | |
|---|---|---|---|---|---|---|
| | SSP1 | SSP3 | SSP5 | SSP1 | SSP3 | SSP5 |
| East Asia | 0.02 [-1.25,-0.03] | <0.01 [-0.16,0.98] | 0.04 [0.01,1.03] | <0.01 [-1.72,-0.72] | >0.33 [-0.36,0.78] | >0.33 [-1.07,0.17] |
| South-East Asia | >0.33 [-0.38,0.06] | <0.01 [0.38,0.84] | <0.01 [0.34,0.82] | <0.01 [-0.78,-0.40] | <0.01 [0.43,0.89] | 0.05 [-0.40,0.00] |
| South Asia | >0.33 [-0.45,0.43] | <0.01 [0.59,1.65] | <0.01 [0.19,1.29] | 0.02 [-0.89,-0.03] | <0.01 [0.81,1.91] | >0.33 [-0.21,0.77] |
| Central Asia | <0.01 [-0.40,-0.10] | <0.01 [0.44,0.86] | 0.01 [0.04,0.50] | <0.01 [-0.52,-0.24] | <0.01 [0.38,0.78] | >0.33 [-0.22,0.10] |
| North America | <0.01 [-2.13,-0.47] | 0.23 [-1.59,0.09] | >0.33 [-1.49,0.23] | <0.01 [-2.19,-0.55] | <0.01 [-1.81,-0.15] | >0.33 [-1.43,0.27] |
| Central America | <0.01 [-0.58,-0.42] | <0.01 [0.23,0.47] | <0.01 [0.49,0.73] | <0.01 [-0.79,-0.63] | <0.01 [0.31,0.59] | <0.01 [-0.35,-0.13] |
| South America | 0.29 [-1.03,0.09] | >0.33 [-0.31,0.89] | >0.33 [-0.38,0.82] | 0.02 [-1.16,-0.04] | >0.33 [-0.26,0.88] | >0.33 [-0.90,0.24] |
| Rus.-Bel.-Uk. | <0.01 [-1.11,-0.31] | >0.33 [-0.64,0.34] | >0.33 [-0.71,0.17] | <0.01 [-1.19,-0.43] | >0.33 [-0.68,0.24] | >0.33 [-0.75,0.15] |
| Europe | <0.01 [-1.36,-0.42] | >0.33 [-0.69,0.29] | >0.33 [-0.68,0.42] | <0.01 [-1.43,-0.51] | >0.33 [-0.73,0.23] | >0.33 [-0.55,0.45] |
| North Africa | <0.01 [-0.56,-0.42] | <0.01 [0.26,0.44] | <0.01 [0.10,0.28] | <0.01 [-0.71,-0.57] | <0.01 [0.29,0.49] | <0.01 [-0.19,-0.03] |
| Sub-Saharan Africa | 0.18 [-0.01,0.13] | <0.01 [0.38,0.60] | <0.01 [0.68,0.88] | <0.01 [-0.30,-0.18] | <0.01 [0.83,1.03] | <0.01 [0.40,0.74] |
| Middle East | <0.01 [-0.48,-0.18] | <0.01 [0.45,0.81] | >0.33 [-0.14,0.34] | <0.01 [-0.75,-0.53] | <0.01 [0.53,0.83] | >0.33 [-0.20,0.1] |

[Figure]

5

**Figure S3: p-values for the ANOVA (Figure 8) corresponding to the factors: (a) Emission Policies (EP), (b) Radiative Forcing (RF), and (c) Interaction (I).**